# CRISPR/Cas9 editing of NKG2A improves the efficacy of primary CD33-directed chimeric antigen receptor natural killer cells

Tobias Bexte [1,2,3,4,20], Nawid Albinger[1,2,3,20], Ahmad Al Ajami[3,5,6], Philipp Wendel [1,2,3,7,8,9], Leon Buchinger[1,2,3], Alec Gessner[3,10], Jamal Alzubi [11,12], Vinzenz Särchen [13], Meike Vogler [13], Hadeer Mohamed Rasheed [14,15], Beate Anahita Jung[14], Sebastian Wolf[3,8,10], Raj Bhayadia [2,3], Thomas Oellerich[3,8,10], Jan-Henning Klusmann [2,3], Olaf Penack [14,16], Nina Möker[17], Toni Cathomen [11,12,18], Michael A. Rieger [3,10,19], Katharina Imkeller[3,5,6] & Evelyn Ullrich [1,2,3,8] ✉

Chimeric antigen receptor (CAR)-modified natural killer (NK) cells show antileukemic activity against acute myeloid leukemia (AML) in vivo. However, NK cell-mediated tumor killing is often impaired by the interaction between human leukocyte antigen (HLA)-E and the inhibitory receptor, NKG2A. Here, we describe a strategy that overcomes CAR-NK cell inhibition mediated by the HLA-E-NKG2A immune checkpoint. We generate CD33-specific, AML-targeted CAR-NK cells (CAR33) combined with CRISPR/Cas9-based gene disruption of the NKG2A-encoding *KLRC1* gene. Using single-cell multi-omics analyses, we identified transcriptional features of activation and maturation in CAR33-*KLRC1*[ko]-NK cells, which are preserved following exposure to AML cells. Moreover, CAR33-*KLRC1*[ko]-NK cells demonstrate potent antileukemic killing activity against AML cell lines and primary blasts in vitro and in vivo. We thus conclude that NKG2A-deficient CAR-NK cells have the potential to bypass immune suppression in AML.

Research on novel chimeric antigen receptor (CAR)-T cell therapies has led to the approval of multiple products for the treatment of B cell-derived malignancies[1–5]. In addition to the U.S. Food and Drug Administration and European Medicines Agency-approved CAR-T cell therapeutics, CD19-targeting CAR-natural killer (NK) cells were reported to be safe and to have similar efficacy as CAR-T cells in a clinical trial[6,7]. The development of CAR products for the treatment of acute myeloid leukemia (AML) is more challenging due to the disease heterogeneity and lack of AML-exclusive antigens[8,9]. Nevertheless, one promising antigen for CAR-based cell therapy is CD33, which is expressed on leukemic blasts and leukemia-inducing cells in up to 88% of AML patients[9,10], and preclinical results have demonstrated anti-AML activity of CD33-specific CAR-T cells[11,12]. One promising alternative for CAR-T cell therapy is adoptively transferred NK cells, which

possess a shorter lifetime than T cells, are associated with fewer side effects, and hold an intrinsic CAR-independent killing capacity, which has been demonstrated in the context of NKG2D-ligand-expressing AML[13–18].

In recent preclinical work, we generated primary CD33-directed CAR (CAR33)-NK cells which showed increased antileukemic activity against CD33[+] AML cells in a xenograft model in vivo[19,20]. However, CAR33-NK cell efficacy can be limited due to the upregulation of immune checkpoints in an immunosuppressive tumor microenvironment. In fact, NK cell activation is tightly regulated by a balance of activating and inhibitory signals mediated by a repertoire of different receptors expressed on the cell surface. Among these, one important inhibitory receptor is NK group 2A (NKG2A) which is encoded by the *KLRC1* gene. It binds to the non-classical MHC I protein HLA-E, which is

often overexpressed on malignant cells[21–25]. We and others have shown that disrupting the NKG2A-HLA-E axis using NKG2A-blocking antibodies, such as monalizumab or CRISPR/Cas9-based disruption of *KLRC1* in NK cells, can significantly enhance NK cell cytotoxicity against multiple myeloma[23–25]. We further observed that various AML cell lines, and also primary AML blasts express high levels of HLA-E, and that HLA-E expression is upregulated following exposure to IFN-γ[25], a cytokine known to be highly secreted by CAR-NK cells following contact with target cells. We therefore hypothesized that the NKG2A-HLA-E axis might play a crucial role in CAR-NK cell exhaustion and that knocking out the NKG2A-coding gene *KLRC1* in CD33-directed CAR-NK would improve their anti-AML killing activity.

In this work, we describe the generation of dual-modified NK cells by combining lentiviral CAR33 transduction with CRISPR/Cas9-mediated *KLRC1* gene disruption. The resulting CAR33-*KLRC1*^ko-NK cells showed significantly increased cytotoxic capacity compared to CAR33-NK or *KLRC1*^ko-NK cells against the CD33⁺/HLA-E⁺ cell line OCI-AML2 and a cohort of patient-derived primary blast cells.

Single-cell transcriptome and epitope measurements revealed activated, mature and cytotoxicity-mediating gene expression patterns in CAR33-NK and dual-modified CAR33-*KLRC1*^ko-NK cells, in accordance with an increased killing capacity. Interestingly, CAR33-*KLRC1*^ko-NK cells displayed increased IFN-γ secretion compared to single-modified *KLRC1*^ko-NK and CAR33-NK cells, which was maintained following target cell contact.

Our findings highlight the clinical potential of altering NK cell surface receptor composition by combining CAR-NK engineering with genome editing to remove inhibitory receptors[26]. Advanced modifications open an array of possibilities and should be considered when designing immune cell-based therapies to target diseases with unmet medical needs.

## Results

### Dual genetically engineered CAR33-*KLRC1*^ko-NK cells combine CAR insertion with reduced NKG2A expression

CAR33-NK cells were generated by lentivirus-based transduction of primary NK cells isolated from peripheral blood (PB) of healthy donors (Supplementary Fig. S1a) to express a second-generation CAR targeting CD33 as shown in Fig. 1a[19]. Following the transduction procedure, CAR expression was achieved in 30-70% of NK cells (Fig. 1b, c). For NKG2A-knockout, non-transduced (NT)-NK and CAR33-NK cells were nucleofected with CRISPR/Cas9 ribonucleoprotein (RNP) complex targeting the *KLRC1* locus. Non-viral RNP transfer resulted in a significant reduction of NKG2A-positive cells by ~50% compared to NT-NK cells (~90% NKG2A⁺) (Fig. 1b, d). Efficient gene disruption frequency of *KLRC1* was confirmed for edited NK cells in the presence or absence of the CAR, by genotyping using the T7 endonuclease 1 (T7E1; >67%) and quantification of insertion/deletion (indels) decomposition by Inference of CRISPR Edits (ICE; >90%) analyses (Fig. 1e, f; Supplementary Fig. S1b. The specificity of this *KLRC1*-targeting CRISPR/Cas9 nuclease in NK cells has been shown previously[25]. Indel distribution analysis revealed the predominant +1 insertion followed by −15 deletion across three different donors in the presence or absence of the CAR. The −15 deletion could be explained by the presence of micro-homology sequences surrounding the cleavage site (Fig. 1f, Supplementary Fig. S1c, 1d). Proliferation increased equally in all four conditions over time (NT-NK cells, *KLRC1*^ko-NK cells, CAR33-NK cells and CAR33-CAR-*KLRC1*^ko-NK cells; shown in log-fold expansion rate, Fig. 1g). However, we observed a transient reduction of the cell numbers following electric pulse-based nucleofection to generate the RNP-based gene knockout (Fig. 1g). The reduction in cell count was directly related to the nucleofection procedure on day 9 (date of CRISPR/Cas9 *KLRC1*-KO generation), but all cell preparations recovered and showed similar growth-rates from day 13 onwards (Fig. 1g, Supplementary Fig. S2a–c). Nucleofection with unrelated sgRNAs or empty MOCK nucleofection

led to a similar NK cell proliferation rate compared to sgRNA targeting *KLRC1* in both CAR33-NK cell and NT-NK cell preparations (Supplementary Fig. S2b, c). Importantly, CRISPR/Cas9-editing of *KLRC1* in CAR33-NK cells showed efficient gene editing that led to a strong reduction of NKG2A cell surface expression (Fig. 1b).

### Multi-omics analysis reveals activated cell state of CAR33-(*KLRC1*^ko)-NK cells

To study the impact induced by CAR33 expression and/or *KLRC1* knockout in NK cells, we determined gene and surface marker expression at the single-cell level using amplicon-based cellular indexing of transcriptomes and epitopes by sequencing (CITE-seq). Therefore, the four NK cell pools (NT, *KLRC1*^ko, CAR33 and CAR33-*KLRC1*^ko) from two different donors were sorted by flow cytometry (Fig. 2a) to obtain an almost 100% pure CAR-expressing (for CAR33 and CAR33-*KLRC1*^ko) and/or NKG2A-negative (for *KLRC1*^ko and CAR33-*KLRC1*^ko) NK cell populations. After quality control and filtering of CITE-seq data, a total of 32,908 and 20,197 cells from donors D1 and D2, respectively, were analyzed in a combined computational workflow. We used a statistical model applied to pseudo-bulk aggregated single-cell data to detect overall changes in expression induced by individual or dual genetic modifications. Our statistical model contained four variables and simultaneously estimated the inter-donor variability, the effect of the CAR (Fig. 2b), the effect of the *KLRC1*^ko (Fig. 2c) as well as the synergistic effect of the dual modification (CAR:*KLRC1*^ko, Fig. 2d). Of note, changes which the model accounts to the CAR:*KLRC1*^ko synergism are changes observed in the CAR33-*KLRC1*^ko-NK cells that cannot be explained by a simple combination of CAR and *KLRC1*^ko (Fig. 2d). The strongest alterations in gene and protein expression were observed upon introduction of the CD33-targeting CAR in NK cells (Fig. 2b, Supplementary Fig. S3a). CAR33-transduced cells expressed higher levels of genes encoding MHC class II proteins, such as *HLA-DQA1, HLA-DQB1, HLA-DRA, HLA-DPB1*, and HLA class II histocompatibility antigen gamma chain (*CD74*) compared to non-transduced (NT)-NK cells (Fig. 2b, Supplementary Fig. S3a). Expression of these genes is commonly observed in activated T cells, and their upregulation in our experiments might be linked to induction of TCR or CAR-induced downstream signaling pathways[27]. CAR33-transduced cells also expressed higher transcription levels of maturation and activation marker genes such as FcγRIIIa/CD16, CD70, integrin alpha M (CD11b), and interleukin-2 receptor alpha chain (IL2Rα/CD25), as well as migration markers such as C-C chemokine receptor type 2 (*CCR2*) and C-X-C chemokine receptor type 4 (*CXCR4*) (Fig. 2b, Supplementary Fig. S3a). Genes encoding proteins expressed on immature or inhibited cells such as Zinc Finger Protein 683 (*ZNF683*) were downregulated upon CAR33 introduction compared to NT-NK cell preparations (Fig. 2b, Supplementary Fig. S3a)[28]. Interestingly, the *KLRC1* knockout in NK cells only led to minor changes in gene and protein expression patterns with a moderate upregulation of FcγRIIIa/CD16 and downregulation of *ZNF683* compared to NT-NK cells (Fig. 2c, Supplementary Fig. S3a). Lastly, the combination of CAR33 expression and *KLRC1* knockout did not lead to any additional alterations in gene and surface marker expression when compared to the single modified NK cells (CAR33 or *KLRC1*^ko) (Fig. 2d). Our computational method detected an interaction between CAR33 expression and *KLRC1*^ko, leading to the apparent upregulation of *ZNF683* and chemokine (C-C motif) ligand 1 (*CCL1*), and downregulation of *CCR2* (Fig. 2d, Supplementary Fig. S3a). This, however, could be explained by the fact that the changes induced by CAR33 and *KLRC1*^ko do not add up in the dual genetic modification: For example, both genetic modifications individually lead to downregulation of *ZNF683*, whereas the combination of both modifications does not further reduce *ZNF683* expression. On the protein level, protein expression analysis by Ab-seq and flow cytometric analyses of the various genetically engineered NK cell preparations revealed no changes of typical NK cell markers (Fig. 2e, f).

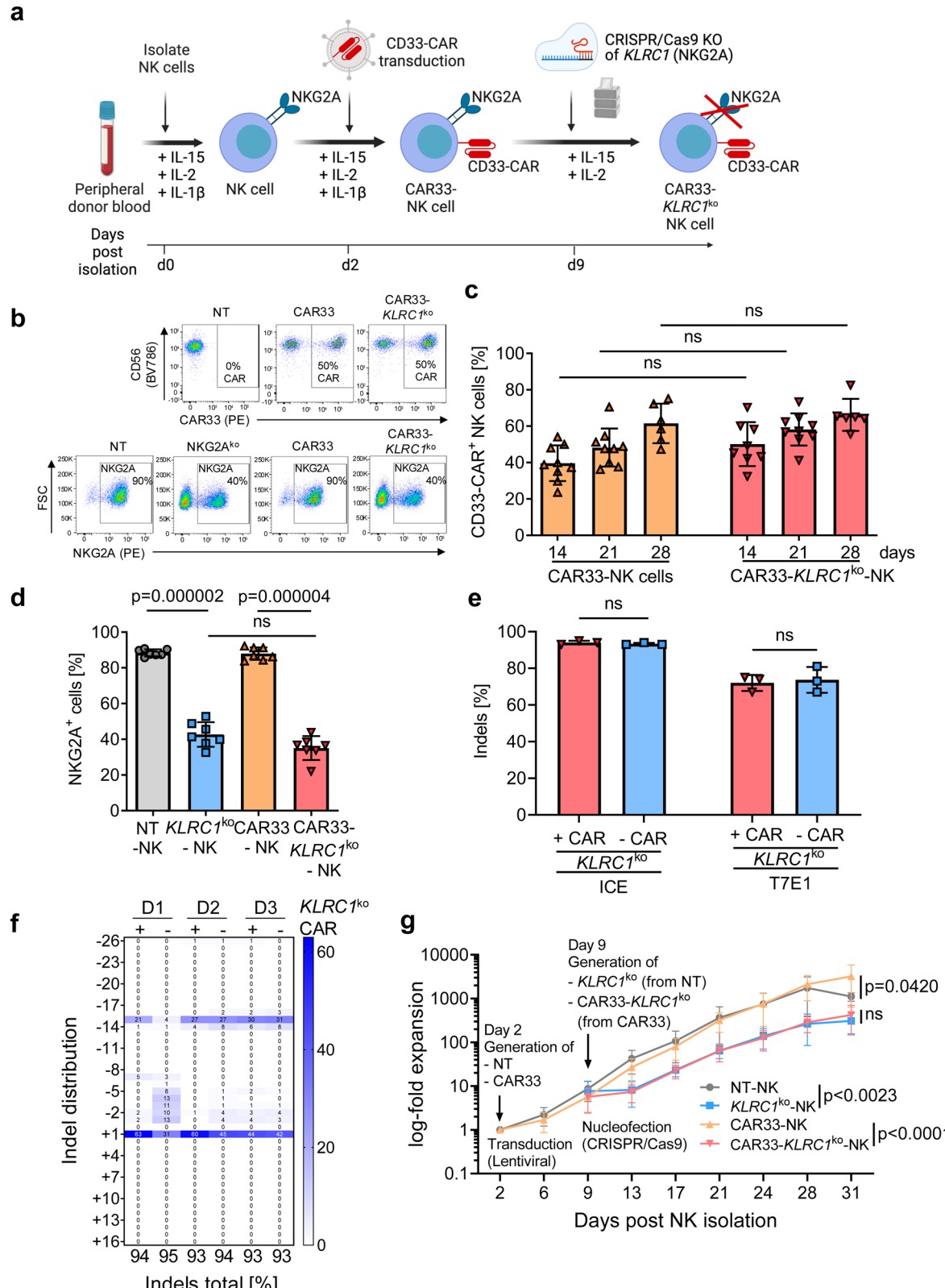

**Fig. 1 | Generation process and evaluation of CRISPR/Cas9 knockout for CAR33-*KLRC1*^ko-NK cells. a** Scheme of the in vitro generation of primary CAR33-*KLRC1*^ko-NK cells. **b** Exemplary flow cytometry plots of CAR33 and NKG2A expression on modified NK cells. **c** Flow cytometry-based CAR33 surface expression analysis over time ($n = 6-9$). Mean ± SD. CAR33 vs CAR33-*KLRC1*^ko: d14 ($n = 0.1158$), d21 ($n = 0.1296$), d28 ($n = 0.8095$). **d** Flow cytometric analysis of NKG2A expression following CRISPR/Cas9 knockout of the *KLRC1* gene ($n = 7$). Mean ± SD. **e** Frequency of *KLRC1* disruption on genomic level was evaluated by Inference of CRISPR Edits (ICE) and T7E1 assay for CAR-transduced (+CAR) and control NK cells (−CAR) ($n = 3$). Mean ± SD. **f** Insertion/deletion (indel) distribution profiles were shown for ICE analysis from three different donors (D1, D2, D3). **g** Expansion-fold and work-flow of non-transduced (NT)-NK, *KLRC1*^ko-NK, CAR33-NK and CAR33-*KLRC1*^ko-NK cells generation in the presence of IL-2 (500 U/mL) and IL-15 (10 ng/ml Miltenyi Biotec or 50 ng/mL CellGenix) ($n = 7$). Mean ± SD (ns = 0.9991). Statistical analysis was performed using two-way ANOVA (**c**, **g**) and paired Student's *t* test (**d**, **e**). The entity of $n$ is biological replicates (from different healthy donors) (**c**–**e**, **g**).

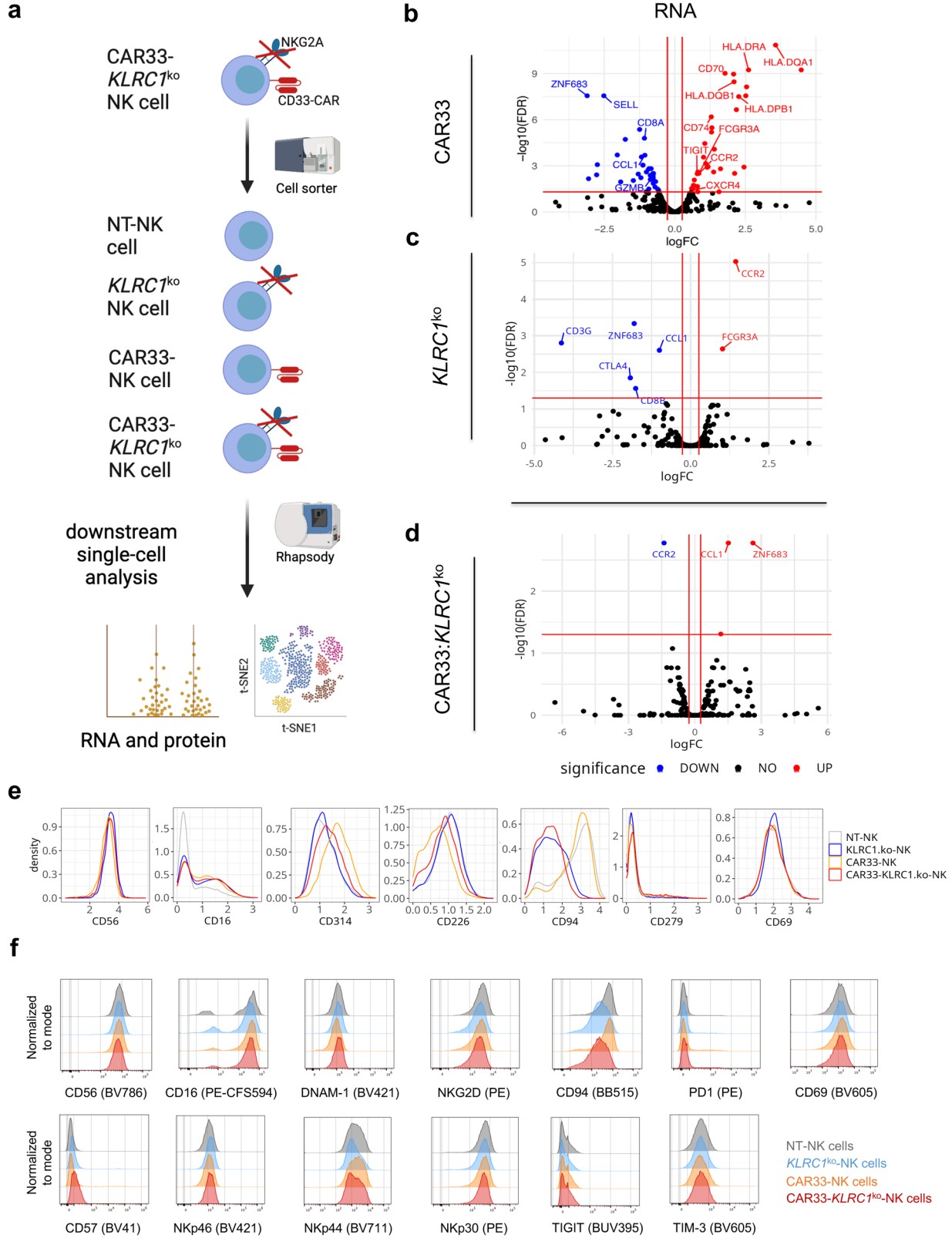

A moderate upregulation of CD16 was confirmed in Ab-seq and cytometric analyses as well as constant phenotype of relevant NK cell receptors like CD56, DNAM1 (CD314), NKG2D (CD226), CD57, NKp44, NKp30 (Fig. 2e, f). Also, no differences were observed for CD69, TIGIT or TIM-3 expression between the four NK cell preparations (Fig. 2e, f).

We next analyzed the combined RNA and protein expression profiles at single-cell level to investigate whether the observed overall expression changes were linked to specific subsets of cells (Fig. 3). Both, the RNA and protein expression on single cell level confirmed the pure NK cell phenotype and absence of T cell markers (see Supplementary Fig. S3b, c). Notably, the composition of the *KLRC1*ko-NK cell pool resembled that of the NT-NK pool and the composition of the CAR33-NK pool was more comparable to that of the CAR33-*KLRC1*ko-NK cell pool (Fig. 3a–c). The pooled sub-clustering of NK cell expression

**Fig. 2 | CAR33 expression and *KLRC1* knockout induce changes in NK cell gene and but not in surface marker expression profiles. a** Scheme of the sorting strategy for NK single-cell analysis of the four different NK cells preparations. CAR-NK cells were sorted on CAR+, *KLRC1*^ko^-NK cells on NKG2A- and CAR33-*KLRC1*^ko^-NK cells on CD33+/NKG2A− NK cells prior to CITE-Seq analysis. **b-d** Overall transcriptional (RNA) changes in NK cells upon introduction of CAR33 (**b**), knockout of *KLRC1* (**c**) and synergistic effects of CAR33 and knockout of *KLRC1* (**d**). Statistical test: quasi-likelihood (QL) F-test against threshold (**b–d**). Volcano plots show combined analysis of NK cells from donors D1 and D2 and indicate adjusted *p*-value (*y*-axis) and inferred logFC (*x*-axis) for each gene feature and for each genetic

modification (CAR33 (**b**), KLRC1^ko^ (**c**), synergistic effect of CAR33 and *KLRC1*^ko^ (**d**)). Up- and down-regulated features are highlighted in color. The horizontal line indicated an FDR of 5% and the vertical lines delimit the range of logFC between −log2 (1.2) and log2 (1.2). Selected genes of interest are labeled. Statistical test: quasi-likelihood (QL) F-test against threshold. **e** Distribution of surface marker expression from CITE-seq data for the four cell preparations. Distributions are displayed as density, with an area under the curve normalized to 1. **f** Surface expression of different receptors of NK cells measured by flow cytometry. Representative histograms are shown.

profiles from all four cell preparations identified six clusters of cells with distinct transcriptional states (Fig. 3c–e, Supplementary Fig. S3d). In order to functionally describe the different transcriptional states, we grouped selected gene or protein features according to their function in "*Checkpoint & Suppressive Regulators*", "*Downstream TCR signaling*", "*Lymphocyte Activation*", or "*Mature NK Cells*" (Fig. 3f, g). The NT-NK cell preparation consisted mostly of cells from transcriptional cluster 3, which was characterized by the expression of genes related to lower levels of NK cell activation and maturation (Fig. 2e–g, Supplementary Fig. S3d). The *KLRC1*^ko^-NK cells showed high proportions of cells belonging to transcriptional cluster 4, which consisted of NK cell genes related to maturation (Fig. 2e–g, Supplementary Fig. S3d). In contrast, the profile of CAR33-NK cells was predominantly related to the transcriptional clusters 1, 2, and 6, in which cells possessed features of activated and matured NK cells and TCR-related downstream signaling, but also of immune suppressive checkpoint pathways (Fig. 3e–g, Supplementary Fig. S3d). Concerning CAR33-*KLRC1*^ko^-NK cells, the transcriptional profile can be classified similar to CAR33-NK cells related to activation and TCR downstream signaling (clusters 1, 2 and 6).

### Knockout of immune checkpoint NKG2A boosts anti-AML activity of CAR33-NK cells in vitro

To assess whether the reduction of NKG2A expression on the surface of CAR33-NK cells, which mostly maintained their activated and mature state (Figs. 2, 3), would lead to increased cytotoxic capacity of CAR33-*KLRC1*^ko^-NK cells, we analyzed their killing capacity against HLA-E and CD33 expressing AML cells, such as the AML cell line OCI-AML2 and patient-derived primary blast cells (Supplementary Fig. S4a and S5). In fact, CAR33-*KLRC1*^ko^-NK cells showed a superior cytotoxic capacity at different effector-to-target (E:T) ratios, compared to NT-, *KLRC1*^ko^- or CAR33-NK cells in both short (4 h) and overnight (24 h) flow cytometry-based killing assays (Fig. 4a) as well as in longitudinal live-cell analyses (Fig. 4b, c). The increased killing capacity of CAR33-*KLRC1*^ko^-NK cells was further illustrated in a 24-h IncuCyte observation, in which the co-incubation of CAR33-*KLRC1*^ko^-NK cells with GFP⁺ OCI-AML2 leukemic cells (green) led to the formation of apoptotic clusters of Annexin V-stained leukemic cells (red), which were significantly less abundant when OCI-AML2 cells were challenged with NT-NK cells (representative images Fig. 4c, Supplementary Movies 1–5). To investigate whether the activated state of CAR33- and CAR33-*KLRC1*^ko^-NK cells was associated with increased perforin/granzyme production and the induction of apoptotic signaling in the target cells, western blot analyses of caspases in survived and sorted OCI-AML2 cells following NK cell contact was performed. We found increased Caspase-3 and Caspase-9 cleavage in OCI-AML2 cells which survived CAR33- and CAR33-*KLRC1*^ko^-NK cell encounters, indicating induction of caspase-mediated cell death (Fig. 4d, Supplementary Fig. S4a, b)[29]. Congenial qPCR analysis of NK cells revealed increased expression of granzyme B in CAR33- and CAR33-*KLRC1*^ko^-NK cells following 2-h incubation with OCI-AML2 cells (Fig. 4e). *KLRC1*^ko^- and CAR33-*KLRC1*^ko^-NK cells were sorted for NKG2A⁻ cells to assess whether they would exhibit increased killing capacity. As a control, *KLRC1*^ko^- and CAR33-*KLRC1*^ko^-NK cells were sorted on CD56⁺ cells to exclude sort-dependent effects. In both

4 h and 24 h cytotoxicity assays, sorted NKG2A⁻, *KLRC1*^ko^- and CAR33-*KLRC1*^ko^-NK cells did not show increased killing of OCI-AML2 cells, compared to non-sorted *KLRC1*^ko^-NK cell population with approximately 50% NKG2A⁻ cells, indicating that a 50% reduction of NKG2A-expression was sufficient to lever out the inhibitory NKG2A-HLA-E axis (Supplementary Fig. S4c).

### HLA-E is a homogenously expressed target in AML patients and CAR33-*KLRC1*^ko^-NK cells demonstrated superior killing capacity against patient-derived blasts cells ex vivo

To further address whether NKG2A knockout CAR33-NK cells had improved efficacy against primary patient material, bone marrow cells (BMCs) derived from ten different AML patients with a substantial blast infiltration (>90% CD45^dim^ AML blasts) were co-cultured for 4 h with CAR33-*KLRC1*^ko^-NK cells generated from different healthy donors (in total *n* = 20 different donor preparations, Fig. 5a). We observed that HLA-E is largely homogeneously expressed in AML patients without significant correlation between different subgroups (cohort *n* = 177, Fig. 5b–d)[30]. HLA-E expressions did not predict overall survival probability (Supplementary Fig. S5a). All of the AML patient-derived BMCs were confirmed by flow cytometry for CD33⁺ positivity and dim to high HLA-E⁺ (representative donor shown in Fig. 5e; gating strategy Supplementary Fig. S5b).

CAR33-*KLRC1*^ko^-NK cells showed improved killing for all conditions and patients compared to NT-, *KLRC1*^ko^- or CAR33-NK cells against AML blast cells from all patients (Fig. 5f, g, total *n* = 10). This holds also true for blast cells derived from patients with high-risk molecular subsets (Fig. 5g, Supplementary Table 1, *n* = 6).

### CITE-seq analysis of CAR33-*KLRC1*^ko^-NK cells revealed preserved activation state and reduced cell cycle activity following AML contact

Our previous results illustrate that the introduction of CAR33 in combination with the knockout of the inhibitory receptor NKG2A synergistically improves the killing capacity of NK cells without inducing strong transcriptional or surface marker expression changes in the respective cell pools. To investigate whether the improved antileukemic killing capacity might be reflected in the phenotypic changes of NK cell pools after AML cell encounter, we additionally performed CITE-seq analysis of the modified and functional NK cells (cytotoxicity data shown in Supplementary Fig. S4d) following a 2 h co-culture with OCI-AML2 cells (E:T = 1:1). NK cells were separated from GFP⁺ AML cells by flow cytometer-based cell sorting. As previously described for the CITE-seq analyses before co-culture, NT-NK cells were sorted as GFP⁻, *KLRC1*^ko^-NK cells as GFP⁻/NKG2A⁻, CAR33-NK as GFP⁻/CAR⁺- and CAR33-*KLRC1*^ko^-NK cells as GFP⁻/NKG2A⁻/CAR⁺ cells (Fig. 6a). When the NK cell expression data after co-culture were compared with those before co-culture, we observed a common pattern of changes in gene and surface marker expression that was induced in all NK cell pools (Fig. 6b, c). The highest increase in expression upon leukemic cell encounter was observed for cytokines and chemokines, such as *CSF2*, *CCL3*, *CCL4*, and TNF, for regulators of immune responses, such as *IL3*, *IL1RN*, CD69, and CD81, as well as for transcriptional regulators, such as *MYC* (Fig. 6b). The expression levels of *FCGR3A*/CD16 as well as *TNFSF8* were reduced

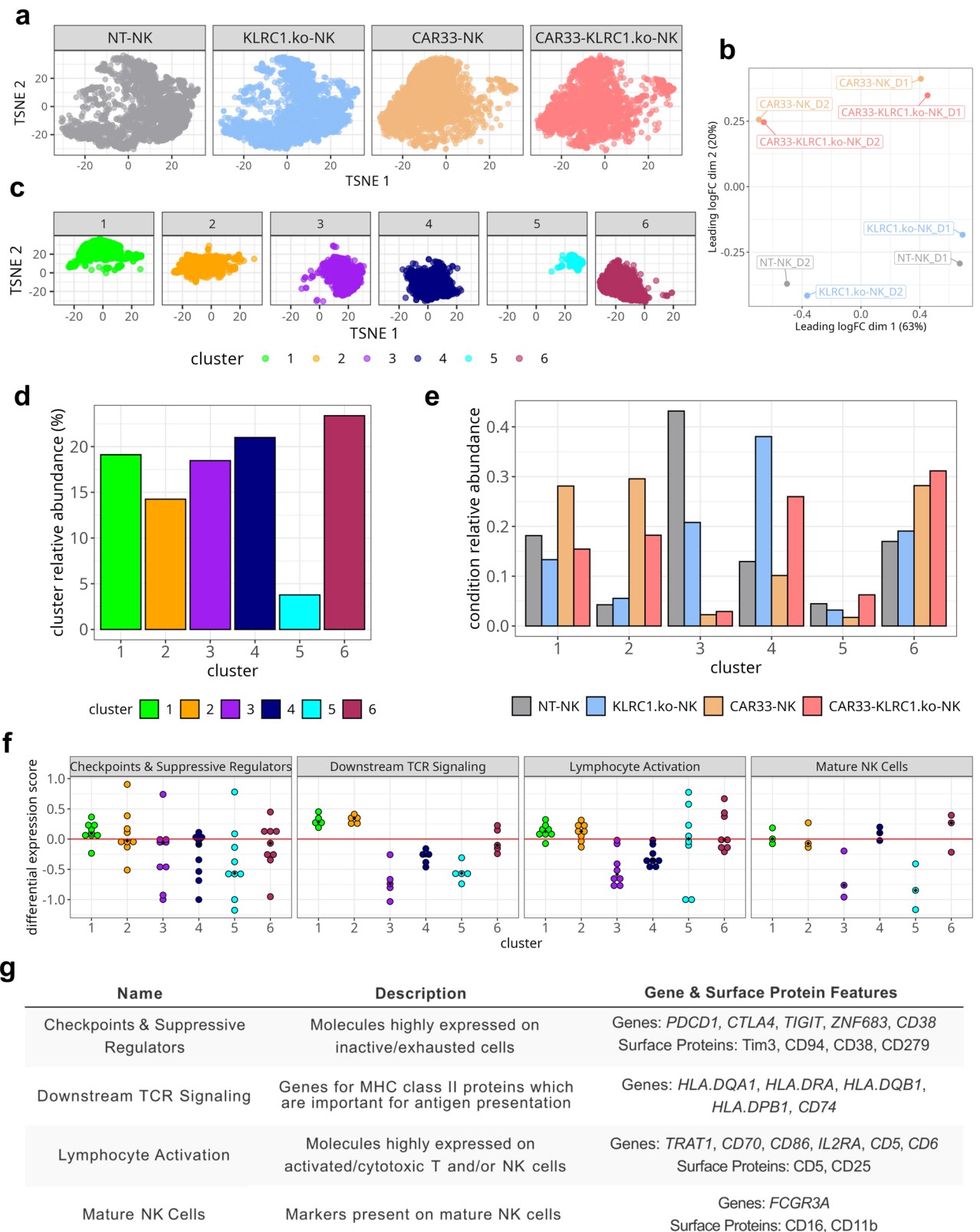

after AML contact in all NK cell pools (Fig. 6b, c). These AML-encounter-induced expression changes were stronger in CAR33 and CAR33-*KLRC1*^ko-NK cells than in NT and *KLRC1*^ko-NK cells, respectively, which is consistent with a higher killing activity of these cell pools (Fig. 6c). CAR33- and CAR33-*KLRC1*^ko-NK cells did not show signs of increased exhaustion through mechanisms such as the upregulation of inhibitory or checkpoint molecules like *HAVRC2* (TIM3), *TIGIT*, *CD96*,

*PD*-1 or *CTLA-4* in line with qPCR analysis following 2 h of co-culture with OCI-AML2 cells (Supplementary Fig. S6a).

We next analyzed the single-cell composition of the different NK cell pools after co-culture with AML cells to study the differences in NK cell-activation induced changes. Similar to what we observed in the NK cell pools before co-culture, the composition of the *KLRC1*^ko-NK pool resembled that of the NT-NK pool and the composition of the CAR33-

**Fig. 3 | CAR33- and CAR33-*KLRC1*^ko-NK cell pools contain an increased fraction of cells with activated, mature NK cell phenotype. a** t-SNE representation of NK cells at single-cell level according to their combined RNA and protein expression profile. Cells are colored and faceted according to the different conditions (NT-NK, *KLRC1*^ko-NK, CAR33-NK and CAR33-*KLRC1*^ko-NK cells). **b** Principal component analysis of differently modified NK cell expression profiles following CITE-seq of two different donors. **c** Pooled subclustering of NK cell expression profiles from all conditions (t-SNE representation as in (**a**)). Louvain clustering on PCA results. **d** Relative abundance of cells belonging to the six clusters as percentage of all sequenced single cells. **e** Relative abundance of cells belonging to the 6 subclusters as fraction of the NK cell pools. For each NK cell pool (NT-NK, *KLRC1*^ko-NK, CAR33-NK and CAR33-*KLRC1*^ko-NK), the relative abundances over all clusters sum up to one. **f** Differential expression score of selected gene or protein features per cluster. The gene and protein features are grouped according to function (Checkpoint & Suppressive Regulators, Downstream TCR signaling, Lymphocyte Activation, Mature NK Cells). Each dot represents one gene or protein feature. The differential expression score of feature X in cluster Y is the mean expression of feature X over all cells in cluster Y divided by its mean over all cells in all other clusters (excluding cluster Y, displayed as log10). A score above zero indicates upregulation of a particular feature in a given cluster compared to the other clusters. Black squares indicate median differential expression score of its features (statistics are medians over genes). **g** Grouping of genes and proteins according to their function. A description of each group and their associated features is shown. **a, c–f** Representative analysis of cells from donor D1.

NK pool was more comparable to the CAR33-*KLRC1*^ko-NK cell pool (Fig. 6d). When these cell samples were clustered according to their gene and surface marker expression profiles, we observed that the transcriptional clusters with the highest levels of co-culture induced genes, namely *CCL4, CCL3, IFNG, TNF*, and *MYC* (Fig. 6d–f, cluster 1 and 4), were enriched in CAR33-NK and CAR33-*KLRC1*^ko-NK cells compared to the other NK cell preparations. A higher abundance of these IFN-γ expressing cells (transcriptional cluster 1), and thus higher IFN-γ levels in supernatants of 4 h co-cultures were observed in CAR33-*KLRC1*^ko-NK cells after co-culture (Fig. 6e–i) compared to CAR33-NK cells, consistent with the finding that this cell pool induced the highest killing capacity.

In addition to the differences in IFN-γ expression, we observed that cells expressing cell-cycle dependent genes (DNA topoisomerase II alpha (*TOP2A*), aurora kinase B (*AURKB*) and *TYMS*, Fig. 6e–h, transcriptional cluster 2) were enriched in CAR33-NK cells that did not carry the *KLRC1* knockout. Despite the fact that a co-culture with AML cells led to an overall decrease in cell-cycle dependent gene expression (Fig. 6h), we were able to experimentally observe a higher proliferation rate of CAR33-NK compared to that of CAR33-*KLRC1*^ko-NK cells following AML cell contact at varying donor-dependent extends (one representative donor is depicted in Supplementary Fig. S6b). On phenotypical level we observed a more activated status after 24 h co-incubation with OCI-AML2 cells for the activation marker CD69 and increased expression levels of DNAM1, NKG2D (Fig. 6j). Higher numbers of NK cells were observed that expressed immune checkpoints like PD-1 and TIM-3, and both CAR33-NK and CAR33-*KLRC1*^ko-NK cells displayed increased TIGIT expression after AML cell contact (Fig. 6j).

## Low numbers of CAR33-*KLRC1*^ko-NK cells can eliminate leukemia-initiating cells located in the bone marrow

After observing a strong anti-AML activity and cytotoxic phenotype of CAR33-*KLRC1*^ko-NK cells in vitro, we aimed to assess the CAR33-*KLRC1*^ko-NK cell activity in vivo (Fig. 7). We used the previously described OCI-AML2-xenograft NOD.Cg-*Prkdc*^scid *Il2rg*^tm1Wjl Tg (CMV-IL3, CSF2, KITLG)1Eav/MloySzJ (NSG-SGM3)-mouse model[19]. NSG-SGM3 mice constitutively produce IL-3, GM-CSF and stem cell factor (SCF), which reflects more physiological conditions of a human host and promotes engraftment of AML cells[31,32]. Initially, we applied 3 times low doses of 5 × 10^6 NK cells to assess whether the increased in vitro cytotoxicity of CAR33-*KLRC1*^ko-NK cells at low E:T ratios could be translated into a preclinical in vivo model. NSG-SGM3 mice received 0.5 × 10^6 OCI-AML2 cells (GFP⁺, Luciferase (Luc)⁺) intravenously, followed by administration of NT-, *KLRC1*^ko-, CAR33-, or CAR33-*KLRC1*^ko-NK cells at day three, seven- and ten post-AML cell injection or remained untreated (UT) (n = 4–9 mice/group; Fig. 7a, b). Treatment with CAR33-*KLRC1*^ko-NK cells resulted in a strong reduction in the leukemic burden by day 9–17, whereas mice that received NT-, *KLRC1*^ko-, or CAR33-NK cells showed only a minor reduction in the AML burden compared to untreated animals (Fig. 7b, c). After the selection and exclusion of two mice per group for bone marrow (BM) re-

transplant experiments, the remaining animals were analyzed in a survival experiment until day 36 (Fig. 7d). CAR33-*KLRC1*^ko-NK cell-treated animals showed a significantly prolonged survival over the CAR33- and *KLRC1*^ko-NK cell-treated animals (Fig. 7d). After these observations of strong antileukemic efficacy in vivo with 3 low doses of 5 × 10^6 NK cells, we further reduced the NK cell doses and injection frequency to 2 times application of 3 × 10^6 CAR33-*KLRC1*^ko-NK cells (Fig. 7e). Again, NSG-SGM3 mice received 0.5 × 10^6 OCI-AML2 cells (GFP⁺, Luc⁺) intravenously, followed by application of NT-, *KLRC1*^ko-, CAR33-, or CAR33-*KLRC1*^ko-NK cells at day three and day ten post-AML cell injection (n = 3–4 mice/group; Fig. 7e). Interestingly, the treatment with CAR33-*KLRC1*^ko-NK cells resulted in a comparably potent reduction of leukemic burden by day 7–17, whereas mice that received NT-, *KLRC1*^ko-, or CAR33-NK cells showed only a minor reduction in the AML burden compared to untreated animals (Fig. 7f, g). Next, we aimed to confirm the absence of AML blast and also leukemia-initiating cells in mice that had no luciferase signal in the bone marrow (Fig. 7h-j). For this purpose, BM-transfer and re-engraftment experiments were performed from a total of 4 CAR33-*KLRC1*^ko-NK cell-treated animals (Fig. 7h) according to the following scheme: BM cells were isolated, pooled and 5 × 10^6 cells were injected intravenously in new mice (NSG-SGM3) per treatment group (Fig. 7i). The BM re-engrafted animals were scored weekly, and the leukemic burden was assessed by bioluminescence imaging (BLI). Mice that received BM from CAR33-*KLRC1*^ko-NK cell-treated animals showed no leukemic cell growth compared to *KLRC1*^ko- and CAR33-NK cell-treated mice (Fig. 7i). In the BM transfer group from *KLRC1*^ko-NK cell-treated mice, one out of six, and from CAR33-*KLRC1*^ko-NK cell-treated animals, five out of five animals stayed free of leukemia until day 120 post-transplantation, when the experiment was terminated (Fig. 7i). Minimal residual disease (MRD) analysis[33,34] of long-term surviving retransplanted mice of the CAR33-*KLRC1*^ko-NK cells group showed negative results in flow cytometry and in confocal microscopy analysis (three exemplary animals, Supplementary Fig. S7a, b). Overall, following both triple and repetitive injections of CAR33-*KLRC1*^ko-NK cells, animals showed no signs of weight loss or changes in appearance or behavior which indicated good tolerability of the applied CAR33-*KLRC1*^ko-NK product. Histological analyses of the liver, lung, and colon tissue revealed no signs of GvHD or other therapy-induced organ damages following double administration of CAR33-*KLRC1*^ko-NK cells (Supplementary Fig. S8a). Moreover, cytokine assessment in mouse serum on day one and day 15 post-NK cell injection revealed low systemic levels of human IL-10, IL-6 production of NSG-SGM3 mice (Supplementary Fig. S8b).

## Discussion

The clinical application of primary CAR-NK cells is a promising cell therapeutic approach for safe and efficient therapies. In Phase I/II clinical trials, allogenic CD19-targeted CAR-NK cells achieved promising response rates in patients with B-cell malignancies, as well as an excellent safety profile with no reported cases of severe cytokine release syndrome (CRS), neurotoxicity, or graft-versus-host disease

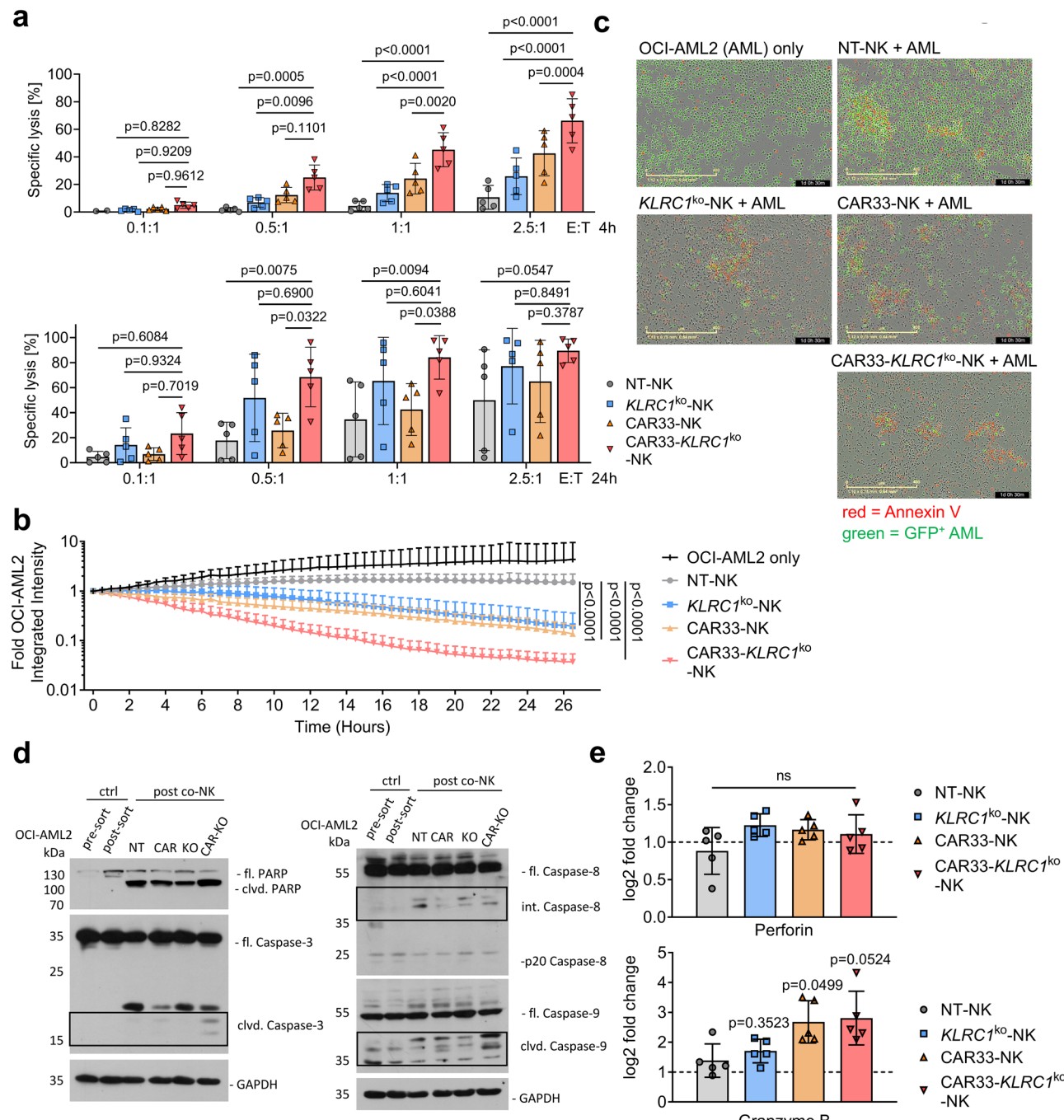

**Fig. 4 | KO of inhibitory receptor NKG2A (*KLRC1*) boosts anti-AML activity of CAR33-*KLRC1*ko-NK cells in vitro. a** Short-term (4 h) and long-term (24 h) flow cytometry-based killing assay of NK cells against CD33+/HLA-E+ OCI-AML2 cells at indicated E:T ratios (*n* = 5). Mean ± SD. **b** Dynamic monitoring of CAR33-*KLRC1*ko-NK-mediated killing using an IncuCyte-S3 imager. CAR33-*KLRC1*ko-NK cells were co-cultured with GFP+ OCI-AML2 cells for 26.5 h and the fluorescence emission was measured over time (OCI-AML2 only condition *n* = 3; rest *n* = 4). Median + range. **c** Representative images taken after 24.5 h of IncuCyte analysis of NT-NK, *KLRC1*ko-NK, CAR33-NK and CAR33-*KLRC1*ko-NK cells co-cultured with GFP+ OCI-AML2 cells (E:T = 0.5:1). Viable tumor cells are shown in green based on their GFP expression.

Apoptotic tumor cells are labeled red by Annexin V staining. For imaging, the IncuCyteS3 platform was used. **d** Caspase-cleavage in survived and sorted OCI-AML2 cells following 2 h NK cell co-culture was analyzed using western blot (one representative experiment of 3 is shown; "clvd" = "cleaved", "ctrl" = "control"). **e** qPCR gene expression analysis of NT-, *KLRC1*ko-, CAR33- and CAR33-*KLRC1*ko-NK cells following 2 h co-cultured with OCI-AML2 cells (E:T = 3:1) (*n* = 5). Mean of technical triplicates ± SD. Statistical analysis was performed by two-way ANOVA (**a**), paired Student's *t* test (**b**), paired Wilcoxon (**e**). The entity of *n* is biological replicates (from different healthy donors) (**a, b, e**).

---

(GvHD)[6,7,35]. The use of NK cells for the generation of CAR effector cells offers potential advantages, particularly in the case of target antigens that are not exclusively restricted to cancer cells, as in the case of AML[36]. In pilot studies, adoptive immunotherapy for AML with unmodified NK cells were confirmed to be safe, with a blast response

observed in some patients[37]. In particular, cytokine-induced NK cell populations seem to have a promising antileukemic potential, with a key role for IL-15 or IL-12/15/18 cytokine-induced NK cell memory[16]. However, the immunosuppressive environment of the leukemic niche in the bone marrow and the resistance of leukemic cells can limit the

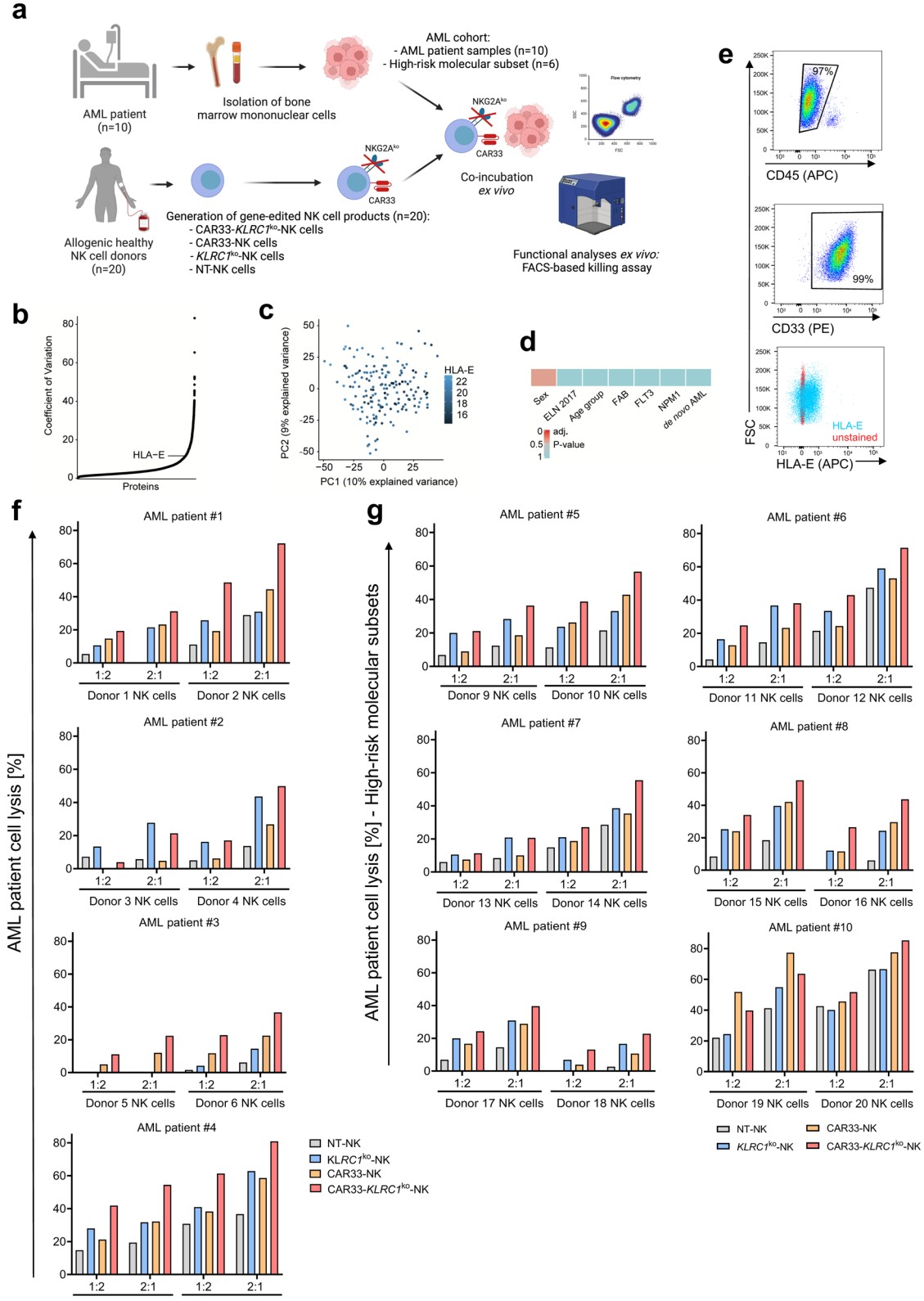

effectiveness of cellular therapeutics. Single-cell analyses have been reported, which revealed the trajectories of NK cell differentiation in the bone marrow and a stress-induced signature in AML, which profoundly impaired NK cell function[38,39]. Importantly, the intrinsic anti-tumor capacity of NK cells has the potential to eliminate also tumor cells that have lost CAR-target antigen expression, which is still not possible in CAR-T cell therapies[11]. Moreover, high levels of secreted

INF-γ can impair the anti-tumor function of NK cells upon tumor contact[24,25]. Elevated levels of secreted INF-γ were also observed for CD33-targeting CAR-NK cells following target cell contact in AML, which can induce upregulation of inhibitory molecules such as the non-classical MHC I protein HLA-E[19,20,24,25]. Consequently the antileu-kemic efficacy can be hampered via inhibitory immune checkpoint receptors such as NKG2A, which bind to HLA-E, often overexpressed

**Fig. 5 | HLA-E is a homogenously expressed target in AML-patients and CAR33-*KLRC1*^ko-NK cells demonstrated superior killing capacity against patient-derived blasts cells ex vivo. a** Scheme of workflow for AML patient preparation and co-incubation with healthy donor NK cells followed by functional read out of lysed patient-derived bone marrow (BM) cells. **b** Rank plot-visualization of coefficients of variation (CV). Every point is a protein's CV ranked by increasing order. HLA-E is highlighted. **c** PCA scatter plot of the first two principal components (PCs) colored by HLA-E intensity. Every point is a sample. **d** Heatmap of adjusted -values from a non-parametric Kruska–Wallis test comparing HLA-E intensity over categories: de novo AML (de novo vs. secondary vs. therapy-related, adj. *p*-value = 1.0), NPM1 (mutated vs. wild type, adj.*p*-value = 1.0), FLT3 (ITD vs. TKD vs. wild type,

adj.*p*-value = 1.0), FAB (M0 to M7, adj.*p*-value = 1.0), Age group (<50 vs. 50-65 vs. >65, adj.*p*-value = 1.0), ELN 2017 (favorable vs. intermediate vs. adverse, adj.*p*-value = 1.0), Sex (female vs. male, adj.*p*-value = 0.37). **e** CD33 expression, HLA-E expression and CD45^dim AML-blast expression of one representative primary BMC sample of AML patient day one post-thawing (one day before co-culture with NK cells) were analyzed by flow cytometry. **f, g** 4 h flow-cytometry-based killing assay of NK cells against primary AML patient material (**f**) and material from patients with high-risk molecular subsets (**g**). Due to partially high rates of spontaneous lysis of patient material post-thawing, the specific killing of viable AML cells is displayed. Mean of technical replicates of AML cell lysis by different NK cell donors (Donor 1-20) are shown for each AML patient #1–10 (**f, g**).

---

on malignant cells[21–23]. Previous data of our group and others showed that HLA-E/NKG2A inhibition could be overcome by blocking or CRISPR/Cas9 NKG2A modification[23–25]. In this work, we describe the successful generation of dual-modified NK cells by combining CAR33 generation with CRISPR/Cas9-mediated *KLRC1* gene disruption. The resulting CAR33-*KLRC1*^ko-NK cells showed superior cytotoxic capacity against CD33^+/HLA-E^+ target cells as compared to either CAR33-NK cells or *KLRC1*^ko-NK cells against AML cell lines, patient-derived primary AML blast cells ex vivo and in a murine xenograft AML model in vivo.

The efficiency of the CD33-CAR expression after lentiviral transduction was in line with previously published data for lentiviral transduction reported by our group and others[11,19,20,40,41]. Notably, the percentage of CAR-positive NK cells increased over time (Fig. 1c), which could be related to the promoter selection[42,43]. The non-viral CRISPR/Cas9 gene disruption frequency of *KLRC1* was highly specific and Indel distribution analysis revealed the predominant +1 insertion followed by −15 deletion in the presence or absence of the CAR and the −15 deletion could be explained by the presence of micro-homology sequences surrounding the cleavage site (Fig. 1f, Supplementary Fig. S1c, 1d). Next-generation sequencing (NGS) analyses indicated only minor activity of the CRISPR/Cas9 nuclease in the CAR-KO NK cells at OT1 (-0.1%) which is similar to previous reports[25] (Supplementary Fig. S1c, d). However, traditional methods such as NGS can only detect unexpected genotoxic events in a limited form of smaller insertions/deletions (INDELs) but do not take into account the entire structural genomic change[42]. New technologies such as CAST-seq allow to detect changes in the entire chromosomal restructuring and should verify the current genome analyses[44,45], even if the risk of mutagenesis is very low in terminally differentiated cells.

Interestingly, although a high frequency of *KLRC1* gene disruption (Fig. 1e) was achieved on the genomic level, this only translated to approximately 50% reduction of NKG2A-positive cells (Fig. 1d). We speculate that the prominent in-frame deletion of −15 nucleotides may have contributed to this effect. The deletion of five residues may abrogate NKG2A function but not necessarily its surface expression. While our work revealed that a 100% NKG2A-negative sorted CAR-NK cell population could not further increase cytotoxicity in vitro (Supplementary Fig. S4c), it remains to be clarified whether such a strong reduction in phenotypic NKG2A expression is necessary in the context of NK cell effector function, which is regulated by the balance of the activating and inhibitory cell surface receptor profile. In this regard, it is important to note that residual NKG2A expression might still hinder CAR-NK cell activity in vivo, due to potential upregulation of HLA-E on tumor cells following contact with IFN-γ secreted by NK cells.

In our previous preclinical work, we generated IL-2/15 activated and expanded primary CAR33-NK cells that showed strong antileukemic activity against CD33^+ AML cells in vitro and in a xenograft model in vivo[19]. High levels of IFN-γ can lead to an upregulation of HLA-E on tumor cells, which has also been observed following CAR33-NK cell therapy in AML-bearing mice as well as in CAR33-*KLRC1*^ko-NK cells following AML cell contact in vitro (Fig. 6g, i)[19,46]. Additionally, the retrospective analyses of AML patient material revealed that HLA-E is

homogeneously expressed on AML cells and does not seem to correlate with clinical response (*n* = 177, Fig. 5b–d, Supplementary Fig. S5a). These observations underline the clinical relevance of the HLA-E/NKG2A axis and highlight the concept of NK cell editing to overcome this immunosuppressive checkpoint. Despite the potentially curative treatment for patients with high-risk acute leukemias by allogeneic hematopoietic stem cell transplantation, disease recurrence remains the major cause of death and alternative therapies are urgently needed[47,48]. In this direction, the CAR33-*KLRC1*^ko-NK cell preparations from various allogenic donors induced a significantly increased cytotoxic efficacy already at low cell numbers against primary AML blast cells (HLA-E^+) derived from a cohort of patients with either standard or high-risk molecular genotype (Fig. 5f, g)[30].

With the goal of further unraveling the underlying mechanisms induced by the specific gene editing, we performed a single-cell transcriptome and surface marker profile analysis of the CRISPR/Cas9-modified CAR33-NK cells. Following direct interaction with AML cells, an increased CAR-mediated downstream signaling profile, and upregulation of the activation and migratory markers were specifically observed in the CAR33- and CAR33-*KLRC1*^ko-NK cell populations, which might explain the enhanced and fast eradication of AML cells. Furthermore, the CITE-Seq analyses revealed a transcriptional profile identified as IFN-γ Type-I-signaling signature predominantly present in the CAR33-*KLRC1*^ko-NK cells after AML exposure. This is in line with recent other studies reporting on IFN-γ activation upon contact with various blood cancer types[39].

Interestingly, when comparing the CAR33-*KLRC1*^ko-NK cell product in the 4 h and 24 h cytotoxicity assays against OCI-AML2 cells, CAR33-NK cells appeared to have predominantly a short-term killing capacity driven by increased caspase-3 cleavage and the release of perforin and granzyme, while *KLRC1*^ko-NK cells developed their increased killing potential in a long-term process (Fig. 4). Importantly, CAR33-*KLRC1*^ko-NK cells benefit from the additional modification, especially at lower E:T ratios, which reflects possible physiological conditions in human (Figs. 5, 7)[6,7,48,49]. While CITE-Seq data underline that CAR33- and CAR33-*KLRC1*^ko-NK cells utilize similar killing mechanisms, the combination of short- and long-term cytotoxicity in dual-modified NK cells appears to be mediated by the CAR and maintained through suppression of the NKG2A checkpoint. This might be crucial for NK cell efficacy in high IFN-γ/HLA-E exposed tumor environment.

Our in vivo experiments do not propose an optimal final dosing but could confirm the potency of dual modified NK cells for the treatment of AML already with two applications of low doses of 3 × 10^6 NK cells and a comparable efficiency after three times 5 × 10^6 NK cells. Clinical data from the recent CAR-NK cell trials for the treatment of B cell malignancies already indicated that strong effects can be achieved with low doses of cell therapy[7]. However, this needs to be further investigated for dual-modified NK cell therapy targeting AML on the way toward clinical translation.

Dual modification of NK cells offers a great potential for advanced, optimized CAR-NK cell products that are also suitable to

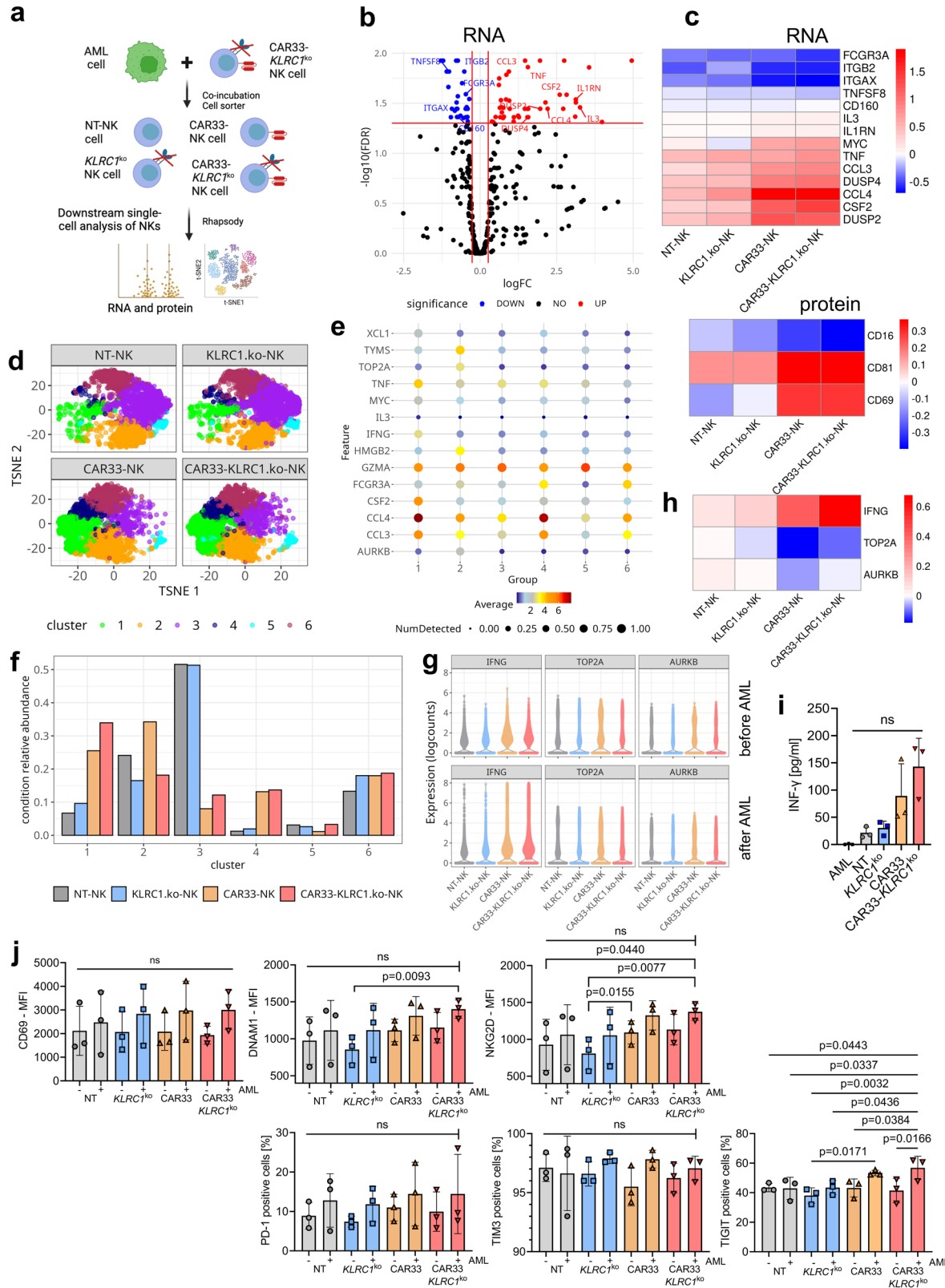

resist to immune suppressive pathways other than NKG2A-mediated. Exemplarily, it has recently been reported that a knockout of the cytokine-inducible Src homology 2–containing (*CISH*) protein in cord-blood-derived CAR-NK cells can improve metabolic "fitness," longer in vivo persistence and cytotoxicity[49]. However, the additional gene editing of immune checkpoints needs to be carefully evaluated in terms of their additional functional efficiency.

Regarding the relevance of the TIGIT receptor-mediated signaling pathway in NK cells, it has been described that deletion of TIGIT could improve the functionality of NT-NK cells but not of CAR-NK cells in AML[50]. In contrast, an optimized effect of dual-modified *TIGIT*-knockout-CAR-NK cells was reported in glioblastoma, which interestingly agrees with our results for the *KLRC1*[KO] of CAR-NK cells in AML[51].

**Fig. 6 | CAR33- and CAR33-*KLRC1*ᵏᵒ-NK cells show preserved activation state and distinct gene expression profiles following AML cell contact. a** Scheme of the sorting strategy for NK single-cell analysis post-co-culture is shown. CAR-NK cells were sorted on CAR⁺, *KLRC1*ᵏᵒ-NK cells on NKG2A⁻ and CAR33-*KLRC1*ᵏᵒ-NK cells on CD33⁺/NKG2A⁻ NK cells following 2 h co-culture with OCI-AML2 cells. **b** Overall transcriptional (RNA) changes in all NK cell pools after co-culture with OCI-AML2 cells. Volcano plot as in Fig. 2**b**–**d. c** Heatmap showing the logarithmic fold change of selected differentially expressed genes and proteins after contact with OCI-AML2 cells compared to before contact in each NK cell pool. A positive logFC represents an upregulation of the gene after co-culture when compared to its expression before co-culture. **d** Pooled subclustering of single cells from all NK cell pools after co-culture (t-SNE representation). Louvain clustering on PCA results. **e** Dot plots illustrating expression of genes and surface markers of interest across the 6 subclusters of NK cells after co-culture with OCI-AML2 cells. Color scale: the unit of measurement is average of log-transformed normalized expression values.

**f** Relative abundance of cells belonging to the 6 subclusters as fraction of the NK cell pools after co-culture. For each NK cell pool (NT-NK, *KLRC1*ᵏᵒ-NK, CAR33-NK and CAR33-*KLRC1*ᵏᵒ -NK), the relative abundances over all clusters sum up to one. **g** Expression values of *IFNG*, *TOP2A* and *AURKB* are compared between the 4 conditions (NT-NK, *KLRC1*ᵏᵒ-NK, CAR33-NK and CAR33-*KLRC1*ᵏᵒ-NK) of different NK cell pools before and after co-culture with OCI-AML2 cells. **h** Heatmap showing the logarithmic fold change of IFNG, TOP2A and AURKB and proteins after contact with OCI-AML2 cells compared to before NK cell contact in each NK cell pool (as in (**c**)). **i** INF-γ levels in supernatants of 4 h co-cultures of NK cells with OCI-AML2 (*n* = 3, biological replicates from different healthy donors). **j** Analyses of NK cell receptor expression after 24 h of co-culture with OCI-AML tumor cells was measured by flow cytometry (*n* = 3 donors). Mean ± SD. (**b**) shows combined data on donors D1 and D2. (**c**–**g**) show the representative analysis of cells from donor D1. Statistical analysis was performed by paired Student's *t* test (**i**) and two-way ANOVA (**j**).

In future clinical translation, the gene editing method is becoming increasingly important, as secondary malignancies have been reported that might have arisen from virally generated CAR-T cell products[52]. In this context, fully non-viral gene editing approaches might reduce the risk of malignant transformation in CAR-immune cell therapies[26,53,54].

Overall, we demonstrated for the first time that CAR33-*KLRC1*ᵏᵒ-NK cells represent an innovative NK cell therapy concept that can overcome AML-mediated immune suppression and appears to be highly functional and safe in preclinical in vivo evaluation. The dual genetic modification by introducing a CAR and precisely knocking out inhibitory checkpoint receptors has the potential to enable NK cells to bypass suppressive effects not only of AML, but possibly also of a broad range of other cancer entities.

## Methods

### Ethics declaration
All studies using human material were approved by the local ethical review board (approval no 329/10 and 274/18) and were performed in accordance with the regulations of Helsinki. The use of bone marrow aspirates was approved by the Ethics Committee of University Hospital Frankfurt (approval No. SHN-12-2016, amend 01 2021 and 02 2022) and the study alliance leukemia (SAL, approval No. EK98032010). All mice experiments were approved by the Regierungspräsidium Darmstadt, Germany (FK1123).

### Primary AML cells
Frozen primary AML patient-derived BMCs were thawed in IMDM medium supplemented with 20% FBS and 1% penicillin/streptomycin and cells were incubated for 10 min with 100 μg/ml DNAse I. Subsequently, cells were washed and cultured in IMDM medium with L-glutamine and HEPES (Invitrogen) supplemented with 100 U/ml penicillin, 100 mg/ml streptomycin, 100 ng/ml stem cell factor (SCF), 20 ng/ml IL-3, 20 ng/ml IL-6 and 100 ng/ml FMS-like tyrosine kinase 3 ligand (FLT3L). Cells were seeded at a cell density of 2×10⁶ cells/ml in a 12-well plate. The next day cells were analyzed for viability via morphology (FSC/SSC) and 7-AAD-staining (BD Biosciences), AML blasts counted based on CD45dim expression using anti-CD45-APC (Clone 2D1, BD Biosciences), CD33 expression using anti-CD33-BV421 (Clone WM53, BD Bioscience) and HLA-E expression using anti-HLA-E-APC (Clone 3D12, Biolegend) antibodies. On day two post-thawing, cells were cultured in a 4-h cytotoxicity assay with NK cells. For detailed patient characteristic see Supplementary Table 1. The use of primary patient material was approved by the Ethics Commission of the University Hospital Frankfurt, Germany (approval no. 274/18). All participants gave written informed consent in accordance with the Declaration of Helsinki.

### Isolation of NK cells
NK cells were isolated from buffy coats of fresh blood from healthy, anonymous donors provided by the German Red Cross Blood Donation (DRK-Blutspendedienst Baden-Württemberg-Hessen, Frankfurt am Main, Germany) as described earlier[55]. In brief, peripheral blood mononuclear cells (PBMCs) were isolated by Ficoll density gradient centrifugation (Biocoll, Biochrom) and NK cell enrichment was performed by utilizing the EasySep Human NK Cell Enrichment Kit (StemCell) according to manufacturer's instructions. NK cell purity was determined by flow cytometry analysis. This project was approved by the Ethics Committee of the Goethe University Frankfurt, Germany (approval no. 329/10).

### Cryopreservation of allogeneic genetically engineered NK cells
Primary NK cells from buffy coats were centrifuged at 300 *g* for ten minutes. The supernatant was removed, and cells were resuspended in cold RPMI1640 medium supplemented with 20% fetal bovine serum (FBS). Subsequently, an equal volume of RPMI1640 medium supplemented with 20% Dimethyl sulfoxide (DMSO) was added dropwise while continuously mixing the cell suspension. The final concentration was hold between (4–16) × 10⁶ cells/ml. The cell suspension was frozen at −80 °C using a Mr. Frosty freezing container (Thermo Fisher Scientific). For long-term storage cells were transferred to liquid nitrogen.

### CAR construction and lentiviral vector production
A second-generation CD33-targeting CAR which incorporates the My96 scFv sequence was utilized as described before[40]. The My96 scFv was coupled to a CD8 hinge and transmembrane domain, 4-1BB/CD137 co-stimulatory domain, and CD3ζ activation domain. A leader peptide including an EF-1α-based promoter derived from GM-CSFRα was added. Self-inactivating baboon envelope-pseudotyped lentiviral vectors (third generation) were manufactured by transient transfection into HEK293T cells using MACSfectin (Miltenyi Biotec) or polyethylenimine (PEI)[56,57].

### Transduction of primary NK cells
NK cells were transduced as described previously[19]. On day two post-NK cell isolation, cells were transduced with lentiviral particles in combination with Vectofusin-1 (Miltenyi Biotec; 2.5 μg/ml final concentration). The virus-cells mixture was centrifuged at 400 *g* for 2 h at 32 °C. 24 h post-transduction half of the medium was replaced by fresh medium. After transduction NK cells were cultured in NK-MACS medium supplemented with 5% human plasma, 1% NK-MACS Supplements, 1% Pen/Strep, 500 IU/ml IL-2 (Miltenyi Biotec or Novartis) and 10 ng/ml IL-15 (Miltenyi Biotec) or 50 ng/ml IL-15 (CellGenix)[19,57].

### *KLRC1*-targeting CRISPR/Cas9 nucleases
Primary NK cells were edited using the non-viral CRISPR/Cas9 system as recently described[25]. Briefly, 1 × 10⁶ pre-cultured (IL-2 + IL-15) non-transduced (NT)-NK or CD33-directed CAR (CAR33)-NK cells were

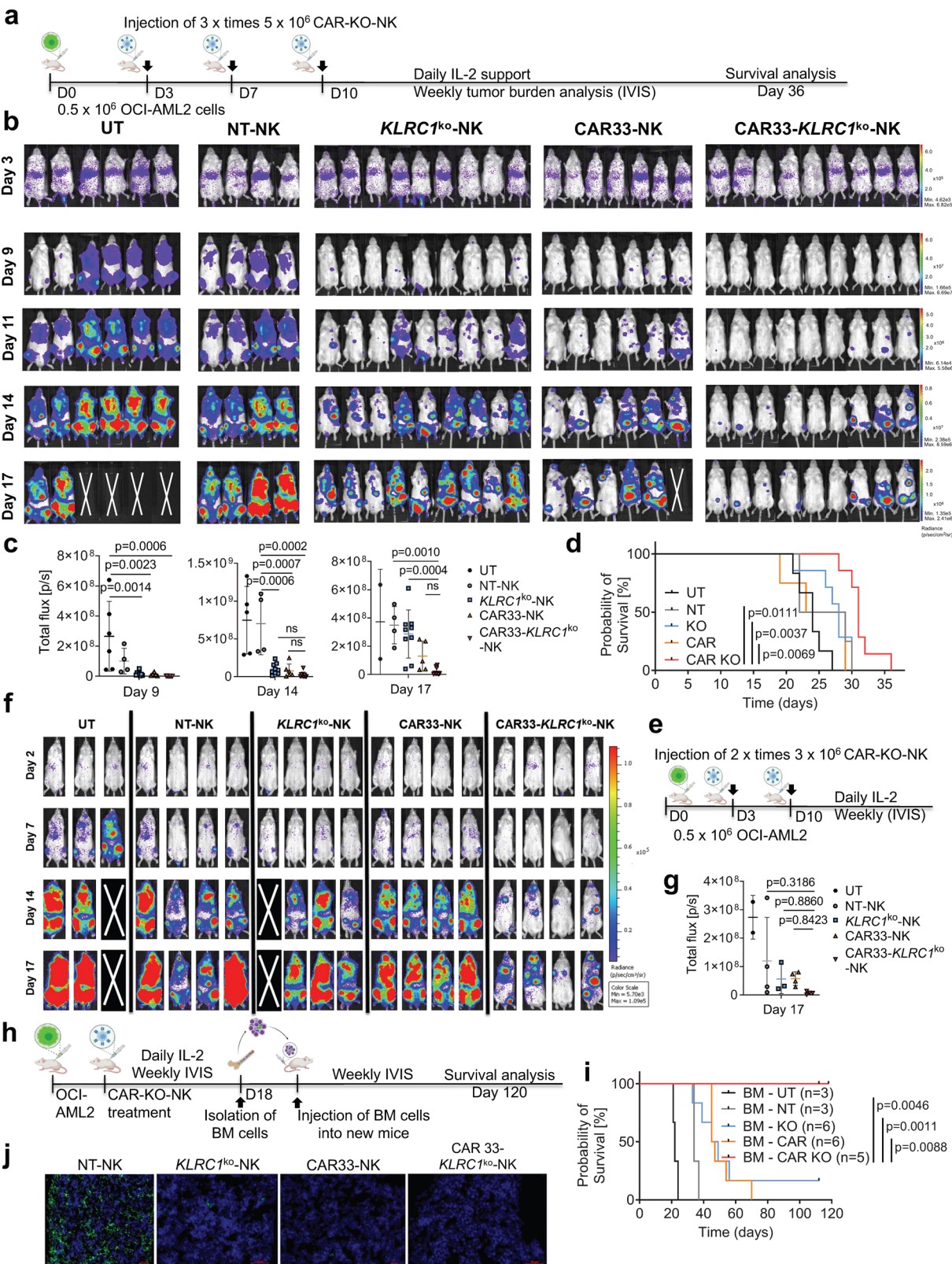

nucleofected (4D-Nucleofector; Lonza) after one week post-transduction or additional culture using amounts of Cas9:gRNA according to the previously optimized NK cell nucleofection protocol[25,53]. Cells were seeded with $1 \times 10^6$ cells/ml in 500 μL culture medium. CAR33-*KLRC1^{KO}*-NK cells and control cells were cultivated for at least 10 days following nucleofection before being deployed in functional assays in vitro or in vivo.

**Genotyping of edited NK cells**

Genotyping of *KLRC1*-edited NK cells was performed using the T7 endonuclease I (T7E1, NEB) and by Inference of CRISPR Edits (ICE) analysis (Synthego Performance Analysis, ICE Analysis. 2019. V3.0). PCR amplicon of 421 bp encompassing target site was generated using primer pair (5′-ctccacctcacccttttaattg-3′ and 5′-caacttggaattctgatctttgc-3′)[25], purified and subjected to T7E1 as previously described, which left

**Fig. 7 | Low doses CAR33-*KLRC1*<sup>ko</sup>-NK cells in an OCI-AML2-xenograft NSG-SGM3 mouse model show improved efficacy compared to *KLRC1*<sup>ko</sup>-NK or CAR33-NK cells.** a Scheme of the in vivo evaluation of a triple injection treatment with CAR33-*KLRC1*<sup>ko</sup>-NK cells (three doses with $5 \times 10^6$ cells intravenously) together with a subcutaneous treatment with IL-2 in OCI-AML2 (Luc⁺) xenograft NSG-SGM3 mouse model. b BLI images of differently treated OCI-AML2 (Luc⁺) engrafted NSG-SGM3 mice over time ($n = 4$–9). Mice received three doses of $5 \times 10^6$ NK cells on day 3, day 7 and day 10 post-AML cell injection. c Total flux analysis (photons/second) mice ($n = 4$–9). Mean ± SD. d Survival and leukemic burden of OCI-AML2 engrafted mice was observed over 36 days. e Scheme of the in vivo evaluation of a double injection treatment with CAR33-*KLRC1*<sup>ko</sup>-NK cells (two doses with $3 \times 10^6$ cells intravenously) together with a subcutaneous treatment with IL-2 in OCI-AML2

(Luc⁺) xenograft NSG-SGM3 mouse model. f BLI images of differently treated OCI-AML2 (Luc⁺) engrafted NSG-SGM3 mice over time ($n = 3$–4). Mice received two doses of $3 \times 10^6$ NK cells on day 3 and day 10 post-AML cell injection. g Total flux analysis (photons/s) mice ($n = 3$–4). Mean ± SD. h Scheme of in vivo bone marrow (BM) re-engraftment experiment. BM cells of NK treated animals (previously untreated (UT), non-transduced (NT)-NK, *KLRC1*<sup>ko</sup>-NK, CAR33-NK and CAR33-*KLRC1*<sup>ko</sup>-NK treated animals) were isolated at day 18, pooled, and injected into new NSG-SGM3 mice ($5 \times 10^6$ cells/animal; $n = 2$–6). i Survival and leukemic burden of BM re-engrafted mice was observed over 120 days (animal protocol: endpoint of the experiment). j Representative confocal microscopy analysis of GFP⁺ AML cells in BM histology d18 post-AML cell injection. Statistical analysis was performed two-way ANOVA (c, g) and Kaplan–Meier (Log-rank (Mantel–Cox) test) (d, i).

228 bp and 193 bp cleaved bands, respectively[58] and/or send for Sanger sequencing to evaluate the indels distribution surrounding the cleaved site by ICE analysis. Our data showed no remaining wild-type sequence in the edited samples as compared to unedited. However, indicating 100% editing efficacy was not possible according to the ICE analysis guideline. Particularly, the ICE analysis is highly dependent on the quality of the Sanger sequencing results, thus the sum of all indel contributions is equal to the R2 value and the remaining sequences defined as unexplained fraction. In-depth evaluation of disrupted alleles was performed by targeted amplicon next-generation sequencing (NGS) using a primer pair spanning the *KLRC1* on-target site (5′-acctgaatctgcccccaaac-3′ and 5′-ccaagctgcacatcctagac-3′)[25]. Paired-end reads were analyzed using the command line version of CRISPResso2.

### Flow cytometry analysis
CD33-CAR and NKG2A expression on gene-modified NK cells as well as the possible contamination with CD3-positive cells were analyzed every 3-7 days post-transduction by flow cytometry using the gating strategy as shown in Supplementary Fig. S1a. The CD33-CAR expression was analyzed by addition of the CD33 protein coupled to biotin and secondary addition of anti-biotin-PE antibody (clone REA746; Miltenyi Biotec). NKG2A cell surface expression was analyzed using an anti-NKG2A-PE antibody (Clone Z199; Beckman Coulter). To detect CD3-positive cell contamination an anti-CD3-BUV395 antibody (clone SK7, BD Biosciences) was used. CD94 cell surface expression was analyzed using an anti-CD94-FITC antibody (Clone REA113, Miltenyi Biotec). Phenotyping protocol was previously published (for gating strategy see Supplementary Fig. S4e)[25,53]. See the full list of all antibodies in Supplementary Table 2.

### CAR-NK cell functional assay
Endpoint cytotoxicity at 4 or 24 h of co-culture induced by either non-transduced (NT)-, *KLRC1*<sup>ko</sup>-, CAR33- or CAR33-*KLRC1*<sup>ko</sup>-NK cells was assessed by flow cytometry. Therefore, target cells either expressing GFP or labeled with Cell Trace CFSE proliferation kit (Invitrogen) were co-cultured with effector cells at various effector-target (E:T) ratios for the indicated time period at 37 °C and 5% CO₂. Afterward, cells were stained with 4′,6-Diamidino-2-phenylindole (DAPI) (AppliChem) and viability of target cells was analyzed using a BD FACSCelesta (BD Biosciences). To analyze the cytotoxicity in real-time, the live-cell imaging IncuCyte S3 system (Sartorius) was used according to the Incucyte Annexin V Dye Product Guide. A 96-well flat bottom plate was precoated with 5 μg/ml fibronectin (Sigma-Aldrich) diluted in 0.1% BSA. To visualize apoptosis in videos, IncuCyte Annexin V Red Dye (Sartorius) was added to the media containing 1 mM CaCl₂ at a final concentration of 2.5 μM. GFP⁺ OCI-AML2 cells were co-cultured with effector cells at E:T ratios of 0.5:1, 1:1 or 2.5:1. Images were acquired every 30 minutes with 10x magnification.

### Cytometric bead array
Cytokine release of NT-, *KLRC1*<sup>ko</sup>-, CAR33- or CAR33-*KLRC1*<sup>ko</sup>-NK cells co-cultured with AML cell lines or primary AML cells, as well as

cytokine levels in blood from the different NK cell-treated NSG-SGM3 mice were determined using a BD Cytometric Bead Array (CBA; BD Biosciences) as described earlier[19].

### Cell preparation for qPCR and western blot analysis
NT-, *KLRC1*<sup>ko</sup>-, CAR33- or CAR33-*KLRC1*<sup>ko</sup>-NK cells were co-cultured with OCI-AML2 cells at an E:T ratio of 3:1 for 2 h in NK-MACS medium supplemented with 5% human plasma, 1% NK-MACS Supplements, 1% Pen/Strep and 500 U/ml IL-2 (Miltenyi Biotec or Novartis). Subsequently, cells were stained with 7-AAD (BD Biosciences) and sorted using a BD FACSAria Fusion device (BD Biosciences). NK cells were sorted on GFP⁻/7-AAD⁻ and OCI-AML2 cells on GFP⁺/7-AAD⁻. Finally, cells were centrifuged, and pellets were frozen at −80 °C for further qPCR or western blot analyses. As a control, OCI-AML2 cells were cultured alone and from each cell type cells were centrifuged, and pellets were frozen at −80 °C without co-culturing.

### Relative mRNA quantification
RNA was isolated using the peqGOLD MicroSpin total RNA kit according to the manufacturer (PEQLAB Biotechnologie GmbH). Isolation of RNA included an on-column DNaseI digestion step (PEQLAB Biotechnologie GmbH). cDNA synthesis was carried out by RevertAid first strand cDNA synthesis kit (ThermoFisher), using an equal amount of 1 μg of RNA for each sample. qRT-PCR was performed on a QuantStudio Flex 7 qRT-PCR cycler (Applied Biosystems) using Sybr Green PCR master mix (Applied Biosystmes). For used primers see Supplementary Table 3. Relative quantification of RNA transcripts was calculated using the ΔΔCq method as described elsewhere[59,60].

### Western blot
Detection of protein expression levels was carried out in whole cell lysates, lysed in Triton-X buffer (30 mM Tris pH [7.4], 150 mM NaCl, 1% Triton-X, 10% glycerol, 0.5 mM PMSF, 2 mM DTT, 1x protease inhibitor cocktail (Roche)). Following antibodies were used for the Western blot detection: rabbit anti-Caspase-3 (Cell Signaling, 9662S), mouse anti-Caspase-8 (Enzo, ADI-AAM-118-E), mouse anti-PARP (poly(ADD)ribose polymerase) (Cell Signaling, 9546S), mouse anti-GAPDH (glyceraldehyde 3-phosphate dehydrogenase) (BioTrend, 5G4-6C5), rabbit anti-Caspase-9 (Cell Signaling, 9502S), goat anti-mouse (Abcam, ab6789), goat anti-rabbit (Abcam, ab6721). Secondary antibodies coupled to horseradish peroxidase, and the signal was acquired using enhanced chemiluminescence detection substrate (ThermoFisher). See the full list of all antibodies in Supplementary Table 2.

### Cell preparation for CITE-seq analysis
For single-cell sequencing analysis, the cryopreserved (genetically modified) NK cells from three different donors were thawed and seeded at $2 \times 10^6$ cells/ml in a 48-well plate in 750 μl in NK-MACS medium supplemented with 1% NK-MACS Supplements, 5% human plasma, 1% Pen/Strep, 500 U/ml IL-2 (Miltenyi Biotec or Novartis) as well as 50 ng/ml IL-15 (CellGenix) per well. Three days later cells were harvested and co-cultured with OCI-AML2 cells in 6 cm dishes at an E:T ratio of 1:1.

OCI-AML2 cells were previously stained with Cell Trace CFSE proliferation kit (Invitrogen) according to the manufacturer's protocol. As a control, NK cells were cultured without contact to AML cells. After 2 h the plated cells were harvested, and an Fc-block was performed using hIgG antibodies (Kiovig and Intratect, University Hospital Frankfurt). Subsequently, cells were stained with an anti-NKG2A-PE antibody (Clone Z199, Beckman Coulter), CD33-CAR-detection reagent (CD33-protein-biotin + anti-biotin-APC (Clone RE746, Miltenyi Biotec) depending on the cell type and each condition was labeled with a different SampleTag from a human Single Cell Sample Multiplexing Kit (BD Biosciences). Subsequently, cells were stained with 7-AAD (BD Biosciences) to exclude dead cells and were sorted using a BD FAC-SAria machine. NK cells were sorted as follows: NT-NK as CFSE⁻, *KLRC1*ᵏᵒ-NK as CFSE⁻/NKG2A⁻, CAR33-NK as CFSE⁻/CAR⁺ and CAR33-*KLRC1*ᵏᵒ-NK cells as CFSE⁻/NKG2A⁻/CAR⁺ cells. Afterwards, single-cell sequencing was performed using a BD Rhapsody Single-cell analysis system (BD Biosciences).

## Single-cell cellular indexing of transcriptomes and epitopes by sequencing (CITE-Seq)

For the single-cell analysis of transcriptomes and epitopes, sorted cells were pooled in a 5 ml FACS tube, to analyze similar cell numbers from each donor and each condition. Cells were washed once and subsequently resuspended in 100 μL Stain Buffer (BSA) (BD Biosciences, cat# 554657). Then, cells were incubated with a master mix containing 2 μL of each of 49 BD AbSeq oligonucleotide-conjugated antibodies (BD Biosciences; see Supplementary Table 4) for 40 min on ice. After washing twice with BD Stain Buffer, cells were resuspended in 620 μL BD sample buffer (BD Biosciences, cartridge reagent kit, cat# 633731). The cell suspension was stained with 3.1 μL 2 mM Calcein AM (BD Biosciences, cat# 564061) and 3.1 μL Draq7 (BD Biosciences, cat# 564904) for 5 min at 37 °C in the dark. Cellular concentration and viability were assessed by using 10 μL of the stained cell suspension in a disposable hemocytometer (Incyto, cat# DHC-N01-5) and the BD Rhapsody Scanner. A cell suspension with 40,000 captured cells was prepared and immediately loaded on the BD Rhapsody cartridge (BD Biosciences, cat# 633733) according to the manufacturer´s protocol. Oligonucleotide-labeled capture beads were loaded on the cartridge and cells were lysed with lysing buffer containing 5 mM DTT. The beads capturing mRNA, Sample Tags and oligonucleotide-labeled antibodies were retrieved and pooled before reverse transcription and exonuclease I treatment were performed (BD Biosciences, cDNA kit, cat# 633773). DNA libraries were prepared using the BD Rhapsody targeted mRNA and AbSeq amplification kit (BD Biosciences, cat# 633774). Therefore, the predesigned human Immune Response Targeted Panel (BD Biosciences, cat# 633750) was used together with a custom-designed primer panel to amplify the cDNA of 434 different genes (see Supplementary Table 5) according to the manufacturer´s protocol. The concentration and size of the final libraries containing indexes for Illumina sequencing were analyzed via a Qubit 4 Fluorometer with a High Sensitivity dsDNA kit (ThermoFisher Scientific, cat# Q32851) and an Agilent 4150 Tapestation with a High Sensitivity D1000 screentape (Agilent, cat# 5067-5584, 5067-5585). Libraries were then pooled to a final concentration of 1 nM and sequenced together with 20 % PhiX DNA in multiple runs (75 bp paired-end) on a NextSeq2000 sequencer (Illumina).

## Single-cell cellular indexing of transcriptomes and epitopes by sequencing (CITE-Seq) data analysis

The generated FASTQ files were uploaded together with panel reference sequences for mRNAs and AbSeqs to the SevenBridges platform (Seven Bridges Genomics, https://www.sevenbridges.com) and processed using the BD Rhapsody analysis pipeline (BD Biosciences) with default settings. Thereby, low-quality read pairs were removed and remaining R1 reads were annotated with cellular barcode and unique

molecular identifier (UMI)s. The R2 reads were then mapped to the reference transcript sequences and reads with identical cell label, UMI sequence and reference gene were collapsed into a single molecule. Afterward, a recursive substitution error correction (RSEC) algorithm was used to adjust molecule counts for sequencing errors. Finally, cell counts were estimated, and Sample Tags were assigned to cells in order to generate RSEC-adjusted molecule count matrices.

## Input data structure

The RSEC-adjusted molecule count matrices of donors D1 and D2 were split into two cells x features count matrices, one for gene expression (RNA) and the other for surface marker (antibody-derived tags, ADT) expression. In total, 60,957 cells were used for the analysis (see SupplementaryTable 6 for summery of the Seven Bridges Genomics platform output).

## Preprocessing workflow

We preprocessed the data by exploring quality control (QC) metrics and filtering cells based on user-defined criteria, by following recommendations by Tomislav Ilicic et al. (2016)[57]. We removed cells assigned as *multiplets* or *undetermined* by the Seven Bridges Genomics platform. Furthermore, we used *scater* [61,62] to remove outliers based on library size and number of features detected. After preprocessing, 32,908 and 20,197 cells from D1 and D2 were retained. Lastly, RNA counts were normalized and logarithmically transformed using *logNormCounts*[61,63] and ADT counts were normalized using the centered log-ratio (CLR) method[63,64], computed independently for each feature.

## Studying the effect of CAR33 and *KLRC1*-KO on NK cells on transcript and surface marker expression before co-culture with AML cells

**Differential gene expression analysis.** Data processing on the two combined datasets of donors D1 and D2 was performed separately on both RNA and ADT count tables. We applied a pseudo-bulk approach, where we summed up feature counts across groups of cells according to condition (NT-NK, *KLRC1*ᵏᵒ-NK, CAR33-NK and CAR33-*KLRC1*ᵏᵒ-NK cells) and donor (D1 and D2). For differential expression analysis of RNA and ADT features, we employed *edgeR*[64] and *limma*[65], respectively.

We designed the model matrix according to the different relevant variables (using the design formula: ~donor + CAR + KO + CAR:KO). For RNA, we fitted a quasi-likelihood negative binomial generalized log-linear model to count data, and tested the differential expression of transcripts relative to a specified fold-change threshold ($=\log2(1.2)$) using *glmQLFit* [64] and *glmTreat* [64]. As for ADT, we transformed count data to log2-counts per million (logCPM) using *voom*[65], and fitted a linear model for each feature using *lmFit* [65]. After that, the empirical Bayes method and the *treat* [61] function where used to test for differential surface marker expression relative to a log2 (1.05) fold-change threshold[66,67]. Finally, for a given contrast, both RNA and ADT features were ranked by *p*-value or absolute logarithmic fold change (logFC) and classified as up- or down-regulated using *decideTests*[65]. We reported our results in volcano (*ggplot2*)[68] and dot plots (*scater*)[61,63] while highlighting particular features of interest.

**Single-cell analysis.** After studying the overall effect of each of the three conditions on a pseudo-bulk level, we also performed a more detailed analysis on the cells before co-culture on single-cell level. The two donors D1 and D2 were analyzed separately.

**Data processing and clustering.** The normalized ADT and RNA feature counts were combined into one feature matrix. Principal component analysis (PCA) was performed with 20 components prior to Louvain clustering[69–72]. Different values for the parameter *k*, which

represents the number of neighbors, were tested and evaluated using clustering similarity scores (adjusted rand index (ARI) & weighted rand index (WRI)). The Louvain clustering result with $k = 50$ returned a reasonable number of clusters and scored highest in both similarity scores (Supplementary Fig. S3c) and was used in downstream analysis. The results were projected on a t-distributed stochastic neighbor embedding (t-SNE)[73].

**Studying the effect of co-culture on transcript and surface marker expression following co-culture with AML**
After studying the effect of *KLRC1*[ko], CAR33 and their interaction together, we wanted to understand the changes of gene and surface marker expression after AML-co-culture. We combined both datasets from D1 and D2 without splitting the cells based on AML-co-culture. This allowed us to compare the cells before co-culture to those after. In total, we had 53,105 cells, of which 29,607 were cocultured with AML-cells. We applied a pseudo-bulk approach, where we summed up feature counts across groups of cells according to before/after co-culture, condition (NT-NK, *KLRC1*[ko]-NK, CAR33-NK and CAR33-*KLRC1*[ko]-NK cells) and donor (D1 and D2).

**Differential gene expression.** We designed our model matrix for the different contrasts using the design formula: ~ coculture + CAR + KO + CAR:KO. From here on, the same procedure explained in the method section "Studying the effect of CAR33 and KLRC1-KO on NK cells" was applied. In addition, we used *scran*'s *find Markers* function to study the differentially expressed features in each condition after co-culture. We calculated the logFC values of RNA and ADT features for each condition after co-culture when compared to the same condition before co-culture. This allowed us to study the effect of co-culture on individual conditions, rather than the general effect that is shown in the previous plots. The results were visualized in a heatmap[68].

**Single-cell analysis.** After understanding the overall effect of co-culture on a pseudo-bulk level, we wanted to study that on a single-cell level. Unlike our pseudo-bulk analysis, we focused on the cells after co-culture in our downstream single-cell analysis and studied both donors D1 and D2 separately to avoid batch-effect issues. This included 18,604 and 11,003 cells from D1 and D2, respectively.

**Data processing.** Similar to the previous single-cell analysis, PCA was computed with 20 components and Louvain clustering was performed on the merged ADT and RNA values using *scran* with $k = 50$, and *igraph*. The results were projected on a t-SNE map. Various summary scores for potential marker features to distinguish between the clusters were computed using *scoreMarkers*[72,73]. The expression values of the top features according to the mean of Area Under the Curve (AUC) of a cluster of interest are visualized in a dot plot across all clusters.

**In vivo functional studies of CAR33-*KLRC1*[ko]-NK cells in xenografted mice**
NOD.Cg-*Prkdc*[scid] *Il2rg*[tm1Wjl] Tg(CMV-IL3,CSF2,KITLG)1Eav/MloySzJ (NSG-SGM3) mice were obtained from *The Jackson Laboratory*, Bar Harbor, ME, USA (JAX stock No.: #013062 (NSG-SGM3)[31]. Mice were held under standardized specific pathogen free/SPF conditions with adequate access to food and water and co-housed experimental/control animals. For an OCI-AML2-xenograft mouse model, $0.5 \times 10^6$ OCI-AML2 (GFP[+], Luc[+]) cells were injected via the tail vein into NSG-SGM3 mice (≤15 weeks old, male). At day two/three post-AML cell application, the leukemic cell engraftment was assessed by injecting luciferin subcutaneously and analyzing the bioluminescence signal using an IVIS Lumina II Multispectral Imaging System (PerkinElmer). On day three and/or seven and day ten $5 \times 10^6$ or $3 \times 10^6$ NT-, *KLRC1*[ko]-, CAR33- or CAR33-*KLRC1*[ko]-NK cells were administered via the tail vein while one group of mice did not receive any treatment.

NK cells were obtained from the same donor and had been cultured for 18 days post-nucleofection at the first time of injection and 25 days post-nucleofection at the time of the next injection. Additionally, on day of NK cells injection a daily subcutaneous injection of 25.000 IU IL-2 (Novartis) was started for mice which received NK cells. The appearance, behavior and weight of the animals were monitored every 2-3 days and the leukemic burden was assessed via BLI, euthanasia was done via cervical dislocation under isoflurane inhalation anesthesia. On day 17 mice were sacrificed or went for survival. For BM re-engraftment analysis BM cells were isolated from each mouse and keept in RPMI1640 with Glutamax, +10% FCS, +1% P/S until i.v. injection. BM cells from each treatment group were pooled and $5 \times 10^6$ cells were injected intravenously in NSG-SGM3 mice (≤16 weeks old, male/female). Mice were routinely weighted and scored and BLI analysis was performed once a week to detect Leukemic growth. Mice were euthanized when BLI values were oversaturated (exposure time of camera saturation of ~60,000 counts), or abortion criteria were reached (weight loss of more than 20% of normal weight, symptoms such as: apathy, noticeable defensive reactions, reduced skin turgor, noticeable breathing difficulties, severe diarrhea, motor abnormalities (e.g., limping, compulsive movements), or an abnormal, unnatural body posture (e.g., hunched posture, opisthotonus). All experiments were approved by the Regierungspräsidium Darmstadt, Germany.

**Statistical analysis**
For statistical analysis, a normal distribution for NK cell functionality was assumed due to the deployment of healthy donors as NK cell source. Concerning the mouse in vivo-experiments also a normal distribution can be assumed. Statistical analysis was performed using GraphPad PRISM version 6-9 (GraphPad Software, Inc.) for experiments with three or more donors/animals.

**Reporting summary**
Further information on research design is available in the Nature Portfolio Reporting Summary linked to this article.

## Data availability
All sequencing data concerning CITE-seq experiments has been deposited in the NCBI GEO repository under accession number GSE221552. Source data are provided with this paper.

## Code availability
All code used to analyze CITE-seq data is available in the following github repository: https://github.com/AGImkeller/CD33_NK_cells_2022.

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

## Acknowledgements

The authors thank all patients who contributed to the study by their donation of leukapheresis samples. The authors thank Ralf Schubert for the support with CBA analysis, and Petra Dinse, Petra Schoen, Bernd Lecher, Simone Maurer, Beate vom Hövel, Daniela Brücher, Katja Stein, Franziska Ganß for their excellent technical assistance, Lea Knapp for experimental support, Sarah Mertlitz for help with creating confocal microscopy images, and the team members from the GSH animal facility and FCI immunomonitoring platform for support of the experiments, and Christian Brandts for support with the UCT Biobank providing AML samples. We thank Lisa Marie Reindl for helpful discussions. Furthermore, we thank the Industrial cooperation partners from BD Bioscience, Sartorius, PerkinElmer, Revvity and Miltenyi Biotec for excellent support in technical applications. This work was supported by the Stiftung Deutsche Krebshilfe (German Cancer Aid) in individual projects (#70114124 to E.U., M.V.) in frame of the Translational Oncology Program (#70114180 to N.A., E.U.) and of the "CAR FACTORY" (#70115200 to E.U., T. Ca.), by the Deutsche Forschungsgemeinschaft (DFG, German Research Foundation) in frame of SFB/IRTG 1292 (Project-ID 318346496 to E.U., N.A., L.B.) and individual projects RI 2462/9-1 and RI 2462/10-1 (to M.A.R), and CA 311/7-1 (to T.Ca.), by the Jose Carreras Leukemia Foundation (to E.U., O.P., M.A.R. by DJCLS 11R/2020, 23R/2021 and DJCLS 15R/2023), by Deutsche Kinderkrebsstiftung DKS_2023_01 (to M.V.), by the foundation "Hilfe für krebskranke Kinder Frankfurt e.V." in frame of the C³OMBAT-AML consortium (to J.-H.K., R.B., E.U.); and by the LOEWE Center Frankfurt Cancer Institute (FCI, Hessen State Ministry for Higher Education, Research and the Arts, III L 5 – 519/03/03.001). T.B. was supported by the DFG, INDEEP Clinician Scientist Program (Project-ID 493624332), T.B., K.I., S.W. and A.A. were supported by the MSNZ Frankfurt (German Cancer Aid), and H.M.R. by the Egyptian Ministry of Higher Education for funding the six-month post-doctoral scholarship in the research group from O.P., Charite University Berlin, Germany. Figures have been created using BioRender CC-BY license (Figs. 1a, 2a, 5a 6a, partly in Figs. 7a, 7e, 7h, and Supplementary Fig. 2a).

## Author contributions

T.B., N.A., L.B., P.W., A.G., V.S., J.A. H.M.R., B.A.J. performed experiments; A.G. preprocessed and A.A., K.I. analyzed the CITE-seq data. V.S., M.V. analyzed the qPCR data. S.W. provided clinical data. N.A., T.B., L.B. analyzed all other data. E.U., K.I., N.A., T.B. designed and directed the study; N.A., T.B., A.A., K.I. performed statistical analysis; E.U., K.I., R.B., T.O., J.-H.K., O.P., M.A.R, T.Ca.; N.M., M.V. discussed the results together with all co-authors. T.B., N.A., K.I., E.U. wrote the manuscript with contribution of all co-authors.

## Funding

## Competing interests

E.U. has a sponsored research project with Gilead and BMS and acts as medical advisor of Phialogics and CRIION. T.O. has disclosures to Merck KGaA: Honoraria; Gilead: Research Funding; Merck KGaA: Research Funding; Roche: Honoraria. O.P. has received honoraria or travel support from Astellas, Gilead, Jazz, MSD, Neovii Biotech, Novartis, Pfizer, and Therakos, he has received research support from Gilead, Incyte, Jazz, Neovii Biotech, and Takeda and is member of advisory boards to Jazz, Gilead, MSD, Omeros, Priothera, Shionogi, and SOBI. N.M. is employee of Miltenyi Biotec. T.B., P.W., E.U. have filed patents on data partly published in this manuscript: PCT/EP2024/060767: Treatment of leukemia with engineered immune checkpoint inactivated CAR-NK cells and CAR-T cells (U31175WO). No competing interests exists for the remaining authors.

## Additional information

[1]Goethe University Frankfurt, Department of Pediatrics, Experimental Immunology and Cell Therapy, Frankfurt am Main, Germany. [2]Goethe University Frankfurt, Department of Pediatrics, Frankfurt am Main, Germany. [3]Goethe University Frankfurt, Frankfurt Cancer Institute, Frankfurt am Main, Germany. [4]German Red Cross Blood Service Baden-Württemberg – Hessen, Institute for Transfusion Medicine and Immunohematology, Frankfurt am Main, Germany. [5]University Cancer Center (UCT), Frankfurt am Main, Germany. [6]Goethe University Frankfurt, University Hospital, Neurological Institute / Edinger Institute, Frankfurt am Main, Germany. [7]Institute for Organic Chemistry and Biochemistry, Technical University of Darmstadt, Darmstadt, Germany. [8]German Cancer Consortium (DKTK) partner site Frankfurt/Mainz, Frankfurt am Main, Germany. [9]German Cancer Research Center (DKFZ), Heidelberg, Germany. [10]Goethe University Frankfurt, University Hospital, Department of Medicine II - Hematology and Oncology, Frankfurt am Main, Germany. [11]Institute for Transfusion Medicine and Gene Therapy, Medical Center – University of Freiburg, Freiburg, Germany. [12]Center for Chronic Immunodeficiency, Faculty of Medicine, University of Freiburg, Freiburg, Germany. [13]Goethe University Frankfurt, Institute for Experimental Pediatric Hematology and Oncology, Frankfurt am Main, Germany. [14]Charité, University Berlin and Humboldt-University Berlin, Department of Hematology, Oncology and Tumor Immunology, Berlin, Germany. [15]Clinical Pathology Department, Faculty of Medicine, Alexandria University, Alexandria, Egypt. [16]German Cancer Consortium (DKTK) partner site Berlin, Berlin, Germany. [17]Miltenyi Biotec B.V. & Co. KG, Bergisch Gladbach, Germany. [18]German Cancer Consortium (DKTK) partner site Freiburg, Freiburg, Germany. [19]Cardio-Pulmonary-Institute, Frankfurt am Main, Germany. [20]These authors contributed equally: Tobias Bexte, Nawid Albinger. ✉e-mail: evelyn@ullrichlab.de

