## [Peer Review File · Nature Communications]

CRISPR/Cas9 editing of NKG2A improves the efficacy of primary CD33-directed chimeric antigen receptor natural killer cellsREVIEWER COMMENTS

Reviewer #1 (expert in NK cells):

In this study, Nawid Albinger and collaborators propose a possible immunotherapy approach for acute myeloid leukemia (AML) based on dual modified CD33 CAR-NK cells engineered for KLRC1 (NKG2A) knock-out by CRISPR/Cas9. CITE-seq analysis before and after AML cells contact reveals that these cells display a more activated phenotype. Accordingly, CAR33 KLRC1ko NK cells exhibit an increased cytolytic activity in vitro against CD33+ HLA-E+ OCI-AML2 cell line and against three AML primary blasts. Finally, the authors show that CAR33 KLRC1ko NK cells have an improved antileukemic activity in vivo in a OCI-AML2-xenograft mouse model.

This work is based on two previous publications of the same research group: the generation of CD33-targeted CAR-NK against AML (Albinger et al, Blood Cancer Journal 2022) and CRISPR-Cas9-based gene editing of NKG2A in NK cells to enhance their cytotoxicity against multiple myeloma (Bexte T et al, Oncoimmunology 2022).

The study is well-written, well-conceived and the experimental controls are appropriate. Although dual modification of primary NK cells could represent a relevant hurdle if this approach is applied to a clinical setting, the findings are very interesting and pave the way to further investigation and improvement. Despite these are important results, there are some issues which should be addressed by the authors.

MAJOR POINTS

1) Data are convincing and some of them confirmed in AML primary blasts, but several other conclusions are based on a single cell line (OCI-AML2). Confirmation with at least another AML cell line would strengthen the results.

2) Potential immunotherapy benefit of these adoptive “double-engineered” NK cells approach depends on the expression of HLA-E at the surface of AML. The authors show HLA-E+ AML blasts derived from 3 patients. Did they have evidence of HLA-E expression in a wider AML cohort?

3) CITE-seq on sorted NK cells allows the authors to identify six cell clusters. Although the authors define some clusters with features of more activated/mature NK cells, their identity is not clearly explained in the Results section.

4) Based on CITE-seq data, engineering NK cells with CAR anti-CD33 alters the NK cell phenotype, inducing a more activated state. These effects seem independent from KLRC1 knock-out and also occur in the absence of CD33+ target cells. The authors should provide possible explanations for this phenomenon.

5) Materials and methods are pretty accurate and detailed. However, the authors could summarize some sections, such as single cell-RNA-seq data analyses, moving details in supplementary material.

MINOR POINTS

6) row 72: Please, fix the error.

- 7) rows 94-96: The authors state that various AML cell lines express HLA-E. However, OCI-AML2 is the only AML cell line showed in supplementary figure 2. Did the authors check other AML cell lines?
- 8) To improve the flow of the reading, the authors should respect a correct progressive figure numbering in the body of the text. (e.g.: row 96: In Introduction section, the authors mention Supplementary figure 2a,c as first figure).
- 9) Supplementary Figure 5a: some lane labels seem unaligned. What is the sample in the last lane on the right?
- 10) Figure 2b – It is difficult to read some gene names (CXCR4, FCGR3A) on volcano plot of CAR33 RNA.
- 11) Figure 2e-f, 5f: Average expression value: please indicate unit of measurement.
- 12) Figure 3b could be enlarged.
- 13) row 601: The authors indicate Table 1 in supplementary data, but I could not find this table.
- 14) Row 216 – Here, data refer to Supplementary Figure 2b. Please, correct.
- 15) Figure 6: Please, in the legend specify in detail the unit of measurement for total flux (“p/s”; I assume “photon/second”).

Reviewer #2 (expert in Adoptive NK cell therapy):

The authors of this manuscript are aiming to enhance CAR-NK cell function in the AML tumor microenvironment by targeting the immunosuppressive receptor NKG2A using CRISPR/Cas9 technology. Combining CAR engineering with CRISPR gene editing to optimize NK cell function has already been done which takes away from the novelty of the approach. This group has previously shown that NKG2A KO in NK cells is beneficial, so the current work is meant to show synergy between NKG2A KO and CAR engineering. While NKG2A is an important inhibitory receptor in NK cells, the authors did not convincingly show on a mechanistic level how NKG2A KO cooperate with the CAR. The transcriptomic and proteomic data show that most of the advantage comes from CAR engineering and not from NKG2A KO which weakens the premise that there could be synergy between the CAR and NKG2A KO.

Major comments:

- Is HLA-E upregulated in primary AML samples? Can the authors show that on the proteomic level by simple flow, and also at the transcriptomic level by looking at publicly available AML datasets?
- Figure 1b: why is the KO efficiency around 50%? RNP mediated CRIPSR editing usually yields >90% efficiency in primary NK cells.
- Figure 1c: it seems like transduction efficiency for the CAR33KO cells is higher than for the WT CAR33NK cells at each time point. This is unexpected since the proliferation rate of the KO cells is lower. Did the authors use a higher MOI for the KO cells? That could contribute to CAR33 KO NK cells having better functionality. The transduction efficiency must be titrated to be the same in order to have a fair comparison between the groups. Can the authors clarify what MOI they used for each condition?
- Figure 2d: what is the comparator in this volcano plot? Is it CAR33 alone without KO? why would ZNF683 and CCL1 be upregulated and CCR2 be downregulated in this case in CAR33KO cells? This trend is opposite to what is seen in NTKO cells when compared to NT. How do the authors explain that?
- Figures 2e and 2f are not well explained. Please elaborate regarding what new information they

bring compared to the volcano plots. If no new information please move to supplementary.

-Figure 3c: How were the 6 subclusters identified? No clear separation is seen in the tsne plots. Can the authors explain how did they delineate the subclusters?

-Line 182-183: "Both KLRC1ko-NK cells and CAR33-KLRC1ko-NK cells were abundant in Cluster 4, which consists of NK cells expressing fewer activation markers." Isn't that against what the authors are trying to convey? the KO is making the NK cells less active? The panel the authors used seems to be T cell centric and not specific to NK cells. What about NK activation markers like DNAM1, NKG2D, NKG2C, Nkp44, Nkp46 and others?

-Line 189-191: "To assess whether the reduction of NKG2A expression on the surface of CAR33-NK cells, which maintained their activated and mature state (Figure 2 and 3), would lead to increased cytotoxic capacity of CAR33-KLRC1ko-NK cells." This statement is not accurate since the authors already showed reduced activation in KO cells as evidenced by decrease in cluster 4.

-Figure 4c and associated supplementary videos: The data shown is misleading since the authors are not showing the important control which is CAR33 alone without KO.

-Figure 4d: Why do CAR NK cells show less cleaved caspase 3 compared to NT counterparts?

-Figure 4e: Seems that the increase in granzyme B is due to the effect of the CAR and not the KO.

-Line 213-216: "sorted NKG2A-, KLRC1ko- and CAR33-KLRC1ko-NK cells did not show increased

killing of OCI-AML2 cells, compared to non-sorted KLRC1ko-NK cell population with approximately 50% NKG2A- cells, indicating that a 50%-KO of NKG2A was sufficient to lever out the inhibitory NKG2A-HLA-E axis." This might be true for an artificial in vitro assay but in vivo having residual NKG2A expression on the NK cells might still hinder their activity. This is especially true given the fact that IFNg can upregulate HLA-E on tumor cells which can enhance NK cell inhibition in vivo.

-Line 217-219: "PBMCs derived from three different AML patients with a substantial blast infiltration (50-95% CD45dim AML blasts) were co-cultivated for 4 hours with CAR33-KLRC1ko-NK cells." Why didn't the authors sort the blast populations and perform the killing assay? Interference from other cells in the PBMCs might muddy the results of the killing assay.

-Figure 5b: Why is CD4 being upregulated in the NK cells? Were these pure NK cells? is there any contamination with T cells?

-CITEseq data in Figure 5d: It seems the effects are mediated by the CAR and the KO is not inducing transcriptional and phenotypic changes that would correlate with higher killing capacity. Do the authors think a longer co-culture time is needed? or phenotyping cells after exposure to tumor in vivo? The data as they are do not support the authors' hypothesis that there is synergy between NKG2A KO and CAR engineering.

-Figure 6a: were the mice also injected with IL-2 or IL-15 to support NK in vivo proliferation and persistence?

-Mice experiments: The in vivo data are weak, only one mouse model was used, the time span of the experiment was short (mice sacrificed at 18 days), the authors used 2 mice only for survival analysis is not appropriate or sufficient to make any conclusions. These data need to be confirmed in a larger group of mice with at least 5 mice per group so that statistics can be performed. Using another mouse model will also strengthen the data. Isolating NK cells from the treated mice and comparing NKG2A KO vs NKG2A WT cells at the transcriptomic and proteomic level would be interesting since from the limited in vivo data it seems that NKG2A KO provides the CAR-NK cells with an advantage in vivo and the effect is more profound than what was seen in vitro.

-Figure 6d: No statistics, visually there doesn't seem to be significant differences between the

different treatment groups. This does not correlate with the BLI images. The CAR33 seems to be performing worse than the other treatment groups based on these graphs.

-Figure 6e: The percentage numbers are so low even for the untreated mice, 2-3% in BM and 0.2-0.5% in spleen? This is odd given the strong BLI signal.

-Figure 6f: The results do not correlate with the BLI images where only CAR33 KO cells seem to be significantly controlling the tumor burden.

-Lines 361-364: "CAR33-KLRC1ko-NK cells showed significantly increased bone marrow infiltration and cytotoxic capacity at lower cell numbers against CD33+/HLA-E+ OCI-AML2 cells compared to CAR33- or KLRC1ko-NK cells in vitro and in vivo without any signs of exhaustion or observable side effects." The authors did not provide data regarding functional exhaustion of the dual modified cells to support this claim. For example, performing long-term tumor rechallenge assays would be very insightful, or even prolonging the in vivo mouse experiments to see if the AML tumor cells eventually come back.

-Lines 317-319: "Additionally, histological analysis of the liver, lung, and colon revealed no signs of GvHD or other therapy induced organ damages following double administration of CAR33-KLRC1ko-NK cells (data not shown)." Please show data, this would be important to ascertain safety of the dual modified cells.

-Lines 413-414: "CAR33-KLRC1ko-NK cells predominantly eliminate AML cells using a Caspase 414 mediated killing mechanism, likely induced by the perforin/ granzyme pathway." The data in Figure 4e show no difference in perforin and the data with increased granzyme b seem to be driven by the CAR.

-Lines 415-417: "Overall, we demonstrated for the first time that CAR33-KLRC1ko-NK cells represent a novel strongly antileukemic NK cell phenotype, which evades exhaustion following AML contact, and appears to be perpetually activated, functional and safe in preclinical in vivo evaluation." Cannot claim that they are perpetually activated, the in vivo model was short, mice were sacrificed at day 18, so we don't really know what happens to the cells after prolonged exposure to AML in vivo and we don't know if they maintain functionality and do not show exhaustion through other mechanisms such as upregulation of TIM3, LAG3, TIGIT or other inhibitory NK receptors

-Many figures are missing statistics, please add that. For example: Figures 1c-f, 4e,f, 5g,h, 6b, 6-h

Minor comments:

-Please number the figures in order. For example Figure 1e coming after figure 1g.

-Figure 1a: please add the timeline of when CAR transduction and CRISPR gene editing were performed from the time of NK cell isolation. Please also indicate how NK cells were activated prior to modifications.

-In the Introduction, please substitute "drugs" by "products" in the following sentence: "Research on novel chimeric antigen receptor (CAR)-T cell therapies led to the approval of multiple drugs for the treatment of B cell-derived malignancies."

-Line 159-160: "Genes encoding proteins expressed on immature or inhibited cells such as Selectin L (SELL) and Zinc Finger Protein 683 (ZNF683)" Please add references for this statement.

-No mention of use of cryopreserved NK cells in the results/figures but this is mentioned in the methods. Please clarify as cryopreservation has a deep impact on NK cell function and phenotype.

-What day after transduction were cells used for functional assays and in vivo assays? Please indicate that in the methods/results.

Reviewer #3 (expert in acute myeloid leukaemia):

This is a well written manuscript in which the authors evaluate a dual modified (CD33 CAR transduced and KLRC1 ko) NK-cells in cell line and xenograft models and demonstrate enhanced anti-AML activity and a more mature and activated transcriptomic profile with such dual modifications that may suggest better efficacy with such NK-CARs in the clinical setting. I have the following critique / comments.

-The authors noted that the proliferation capacity of the KLRC1 KO CD33 NK-CAR was reduced compared with those without KLRC1 KO. Can they explain the reason for this phenomenon? This is somewhat concerning. Would partial KLRC1 KO be feasible and have continued desired effect of avoiding NKG2DA-HLA E axis inhibition while not compromising the NK-CAR proliferative potential?

-In the co-culture experiment when PB blasts from 3 AML patients were co-cultivated for 4 hours with CAR33-KLRC1ko-NK cells please provide the baseline molecular and cytogenetic features as well as status (frontline or R/R) of these three AML populations. Do the authors postulate that a particular molecular or cytogenetic subset of AML would be more susceptible to CAR33-KLRC1ko-NK cell? Potentially based on known percent expression or MFI of CD33 or HLA-E in certain AML subsets? This would help select and enrich for certain populations in the initial phase I studies.

-The authors used the OCI-AML-xenograft model for in vivo assessment and showed that the CAR33-KLRC1ko-NK cells most effectively eradicated bone marrow and organ leukemia in mice. However these in vivo experiments could be much more representative if they were performed in patient derived xenografts encompassing various molecular subsets , especially high risk molecular subsets such as TP53m or inv3q or complex cytogenetics harboring AML. These are the populations of greatest unmet need and if the CAR33-KLRC1ko-NK cells show robust activity in such PDXs this could help focus on these populations in the initial phase I studies allowing for a more rapid and directed clinical development than trying to evaluate in all comer populations,

-What do the authors anticipate will be the minimal effective number of CAR33-KLRC1ko-NK cells for dosing in patients with AML to start seeing efficacy. How frequent dosing would be needed?

-While NK-cell based therapies have shown encouraging early efficacy in both lymphoma and AML the durability of responses has been questionable. What makes the authors believe this dual modified CAR product will have better durability. Is this something they could model in PDXs and demonstrate for example comparing such dual modified NK-cells versus traditional CD33 CART products and assessing not only AML burden clearance but time to relapse of AML in vivo models? Such data would be compelling to suggest the dual modifications implemented will not only help boost the initial cell kill but also potentially the durability of the NK response.

- The discussion repeats much of the results and can be considerably shortened. Please avoid this and in the Discussion only focus on the implications of the key findings described in Results and future directions with this cell therapy approach.

Point-by-point-reply Nature Communications manuscript NCOMMS-23-02540-T

We thank the reviewers for all comments and the supportive feedback on our work, which we greatly appreciated and which led to the following improvement of our manuscript:

- 1) Superiority of the KLRC1KO CD33 CAR NK over the CAR NK cells has been clearly and statistically confirmed in additional experiments *in vivo*. All *in vivo* experiments were carried out on additional mice so that meaningful statistical analyses and survival analyses have been added (**please, see our comments to reviewer 2**).
- 2) All datasets are quantified and statistically analysed where relevant (**please, see our comment to reviewer 2**).
- 3) *In vitro* assays have been confirmed on a large set of primary AML blasts derived from patients to evaluate the killing capacity in addition to our work with the OCI-AML2 cell line (**please see our comments to reviewer 1 and 3**).
- 4) The impact of the genetical modifications on proliferation of the CAR33 KLRC1KO NK compared to all relevant controls has been addressed in detail (**please see our comment to reviewer 2**).

Finally, we addressed all comments from the reviewers (**as stated below in the point-by-point-reply**) and we are convinced that our work has been significantly improved.

We added **1 additional figure** and supplementary information, and highlighted all modifications in the revised version of our manuscript accordingly.

Reviewer #1 (expert in NK cells):

In this study, Nawid Albinger and collaborators propose a possible immunotherapy approach for acute myeloid leukemia (AML) based on dual modified CD33 CAR-NK cells engineered for KLRC1 (NKG2A) knock-out by CRISPR/Cas9. CITE-seq analysis before and after AML cell contact reveals that these cells display a more activated phenotype. Accordingly, CAR33 KLRC1ko NK cells exhibit an increased cytolytic activity *in vitro* against CD33+ HLA-E+ OCI-AML2 cell line and against three AML primary blasts. Finally, the authors show that CAR33 KLRC1ko NK cells have an improved anti-leukemic activity *in vivo* in a OCI-AML2-xenograft mouse model.

This work is based on two previous publications of the same research group: the generation of CD33-targeted CAR-NK against AML (Albinger et al, Blood Cancer Journal 2022) and CRISPR-Cas9-based gene editing of NKG2A in NK cells to enhance their cytotoxicity against multiple myeloma (Bexte T et al, Oncoimmunology 2022).

The study is well-written, well-conceived and the experimental controls are appropriate. Although dual modification of primary NK cells could represent a relevant hurdle if this approach is applied to a clinical setting, the findings are very interesting and pave the way to further investigation and improvement. Despite these are important results, there are some issues which should be addressed by the authors.

Major points:

1) Data are convincing and some of them confirmed in AML primary blasts, but several other conclusions are based on a single cell line (OCI-AML2). Confirmation with at least another AML cell line would strengthen the results.

We thank the reviewer for this feedback and valuable comment. We are pleased to provide additional functional data addressing the cytotoxic capacity targeting AML cells from different sources. In the revised version, we strengthen our functional data of AML primary blasts up to a total number of n=10 patients (low and high-risk molecular subsets) by using NK cell preparations from independent and different healthy donors (n=20). Please see the new **Figure 5** of the revised manuscript and the highlighted addition to the **Results** section.

2) Potential immunotherapy benefit of these adoptive “double-engineered” NK cells approach depends on the expression of HLA-E at the surface of AML. The authors show HLA-E+ AML blasts derived from 3 patients. Did they have evidence of HLA-E expression in a wider AML cohort?

We thank the reviewer for raising this important point. We are pleased to provide the requested HLA-E data of a wide AML cohort. In the revised version, we added data of retrospectively analyzed HLA-E expression in a broad AML patient cohort (n=177) and confirmed that HLA-E is uniformly expressed in all patients. Please see new **Figure 5b,c,d** and **Supplementary Figure S5**, and the highlighted addition to the **Results** section.

3) CITE-seq on sorted NK cells allows the authors to identify six cell clusters. Although the authors define some clusters with features of more activated/mature NK cells, their identity is not clearly explained in the Results section.

We thank the reviewer for this important remark. We have taken up this comment and added a clearer explanation *about the NK cell cluster identity and the computational methods used* in the **Methods** section.

First, it is important to kindly keep in mind that the clusters identified in our analyses do not identify distinct subtypes of NK cells, but rather groups of NK cells with distinct transcriptional states. The reason why we perform clustering is that we want to highlight that the overall up-/downregulation of certain genes shown in **Fig. 3 and Fig. 6** is due to a differential abundance of cells exhibiting a specific transcriptional state. The higher expression of certain groups of genes is restricted to specific subgroups of cells.

Regarding the computational methodology, we selected the parameters for clustering using different metrics. We added the following text to the **Methods** section, to better explain how clustering was performed:

*"The normalized ADT and RNA feature counts were combined into one feature matrix. Principal component analysis (PCA) was performed with 20 components prior to Louvain clustering. Different values for the parameter k , which represents the number of neighbors, were tested and evaluated using clustering similarity scores (adjusted rand index (ARI) & weighted rand index (WRI)). The Louvain clustering result with $k=50$ returned a reasonable number of clusters and scored highest in both similarity scores (**Supplementary Fig. 3c**) and was used in downstream analysis. The results were projected on a t -distributed stochastic neighbor embedding (t -SNE)."*

As described in this addition to the methods section, a figure, which depicts the clustering behavior using different parameter values, is added into the **supplementary figure S3c**.

4) Based on CITE-seq data, engineering NK cells with CAR anti-CD33 alters the NK cell phenotype, inducing a more activated state. These effects seem independent from KLRC1 knock-out and also occur in the absence of CD33+ target cells. The authors should provide possible explanations for this phenomenon.

We thank the reviewer for addressing this question. We have the impression that our part on the interpretation of the CITE-Seq data was not yet precisely enough and could lead to misinterpretations, so we have revised it for more clarity. Finally, the detailed analysis of the CITE-seq data leads to the conclusion that genetic modification towards CAR expression leads to stronger transcriptional changes compared to the KO of KLRC1 (**Fig. 2, 3**) independent of the exposure to the target. At the protein level, the ADT and FACS analyses

show no significant differences in the activation status of the NK cell preparations. This is not to be expected as identical conditions are used for NK cell isolation, cultivation and expansion (**as now also stated in the Discussion part**). However, after contact with the CD33+ target cells, differences in the direction of activation of the CAR-expressing and dual-modified NK cell preparations become apparent, which are reflected in the RNA profile, and then in later FACS analyses (**Fig. 6j**).

5) Materials and methods are pretty accurate and detailed. However, the authors could summarize some sections, such as single cell-RNA-seq data analyses, moving details in supplementary material.

Thank you for this suggestion. We have summarized parts of the single-cell analysis methods to make this section more concise. However, we are not allowed to move parts of the methods to the supplementary section, as commented by the journal editors: *"The comment from reviewer 1 regarding moving some of the Methods to Supplementary Information should be disregarded, as this is not permitted as per journal style."*

Minor points:

6) row 72: Please, fix the error.

Thanks for the comment. However, we could not identify any error in line 72 to which the reviewer refers, but we went completely and carefully through the manuscript as part of the revision in order to correct possible errors.

7) rows 94-96: The authors state that various AML cell lines express HLA-E. However, OCI-AML2 is the only AML cell line showed in supplementary figure 2. Did the authors check other AML cell lines?

We thank the reviewer for this question. We are pleased to inform you that we were able to analyze the expression of HLA-E in a broad AML patient cohort (n=177). As mentioned above (see comment #2), we were able to confirm that HLA-E is homogeneously expressed in the representative and entire AML patient cohort (n=177). These data are shown in the **new Figure 5** and Supplementary Figure 5 and we have also updated this information in the highlighted addition to the **Results** and **Discussion** section.

8) To improve the flow of the reading, the authors should respect a correct progressive figure numbering in the body of the text. (e.g.: row 96: In Introduction section, the authors mention Supplementary figure 2a,c as first figure).

Thank you for the remark. We restructured our Supplementary Figures in a way that they are mentioned more chronologically in the text.

9) Supplementary Figure 5a: some lane labels seem unaligned. What is the sample in the last lane on the right?

Thanks for the remark. It is the positive control "CCR5d32". We included a labling "pos ctr". Furthermore, we aligned the lines and added "NT" instead of the "-" to make clearer that here *KLRC1* was knocked out in the NT-NK cells. Please see the updated **Supplement Figure S1** (former Fig. S5) of the revised version.

10) Figure 2b – It is difficult to read some gene names (CXCR4, FCGR3A) on volcano plot of CAR33 RNA.

Thanks for this suggestion. We have edited **Figure 2b** to be more visually appealing. Please see the new version of the resubmitted work.

11) Figure 2e-f, 5f: Average expression value: please indicate unit of measurement.

Thanks for the comment, we indicated and added the following units of measurement to the **updated Figure legends**:

Figures indicated by the reviewer were moved to **Fig. 6e** and **S3a+d**.
RNA: average of logarithm-transformed normalized expression values
Protein: average of CLR-transformed expression values

12) Figure 3b could be enlarged.

Thank you for this suggestion. We have enlarged **Figure 3b**.

13) row 601: The authors indicate Table 1 in supplementary data, but I could not find this table.

We re-uploaded the corresponding Table 1 in the re-submitted version ("Suppl_Table 1-BD AbSeq")

14) Row 216 – Here, data refer to Supplementary Figure 2b. Please, correct.

Thank you for identifying this mistake. This has been corrected accordingly.

15) Figure 6: Please, in the legend specify in detail the unit of measurement for total flux ("p/s"; I assume "photon/second").

Correct, we specified this in the figure legend.

Reviewer #2 (expert in Adoptive NK cell therapy):

The authors of this manuscript are aiming to enhance CAR-NK cell function in the AML tumor microenvironment by targeting the immunosuppressive receptor NKG2A using CRISPR/Cas9 technology. Combining CAR engineering with CRISPR gene editing to optimize NK cell function has already been done which takes away from the novelty of the approach. This group has previously shown that NKG2A KO in NK cells is beneficial, so the current work is meant to show synergy between NKG2A KO and CAR engineering. While NKG2A is an important inhibitory receptor in NK cells, the authors did not convincingly show on a mechanistic level how NKG2A KO cooperate with the CAR. The transcriptomic and proteomic data show that most of the advantage comes from CAR engineering and not from NKG2A KO which weakens the premise that there could be synergy between the CAR and NKG2A KO.

Major comments:

-Is HLA-E upregulated in primary AML samples? Can the authors show that on the proteomic level by simple flow, and also at the transcriptomic level by looking at publicly available AML datasets?

We thank the reviewer for raising this important point. As mentioned also to reviewer#1: We are please to provide the requested HLA-E dataset of a wide AML patient cohort (n=177) and we also confirmed HLA-E by flow cytometry for all primary AML samples used for in the new version of the manuscript (n=10). In the revised version, we added a dataset of retrospectively analyzed HLA-E expression in a broad AML patient cohort (n=177) and

confirmed that HLA-E is uniformly expressed and does not correlate with clinical response (see **Figure 5b,c,d** and **Supplementary Figure S5**).

We further confirmed HLA-E in primary AML samples of up to n=10 patients by flow cytometry. Please see new **Figure 5e**. Additions in the **Results** and **Discussion** are highlighted in the revised version of the manuscript.

-Figure 1b: why is the KO efficiency around 50%? RNP mediated CRISPR editing usually yields >90% efficiency in primary NK cells.

We thank the reviewer for this comment. In the revised version of the manuscript, we have added NGS results showing 97-99% efficient modification, and we have included a more detailed discussion of that aspect in the **Discussion** section as follows:

*"Interestingly, although a high frequency of KLRC1 gene disruption (**Fig. 1e**) was achieved on the genomic level, this only translated to approximately 50% reduction of NKG2A-positive cells (**Fig. 1d**). We speculate that the prominent in-frame deletion of -15 nucleotides may have contributed to this effect. The deletion of five residues may abrogate NKG2A function but not necessarily its surface expression. However, while our work revealed that a 100% NKG2A-negative sorted CAR-NK cell population could not further increase cytotoxicity in vitro (**Supplementary Fig. S4c**), it remains to be clarified whether such a strong reduction in phenotypic NKG2A expression is necessary in the context of NK cell effector function, which is regulated by the balance of the activating and inhibitory cell surface receptor profile. ..."*

-Figure 1c: it seems like transduction efficiency for the CAR33KO cells is higher than for the WT CAR33NK cells at each time point. This is unexpected since the proliferation rate of the KO cells is lower. Did the authors use a higher MOI for the KO cells? That could contribute to CAR33 KO NK cells having better functionality. The transduction efficiency must be titrated to be the same in order to have a fair comparison between the groups. Can the authors clarify what MOI they used for each condition?

We thank the reviewer for this important question concerning the used MOI. We used MOI of 2 for every experiment and setting (now added to the **Methods** section). In our previous work, where we setup the CAR33 NK cell generation, we identified the MOI2 as optimal for an efficient transduction (Albinger et al, *BCJ* 2022; Albinger et al, *BMT* 2024). For more clarity, we added a scheme of the experimental setting in **Fig. 1a** and improved the graphical layout of **Fig. 1g** concerning cell proliferation during the engineering and manufacturing procedures.

Furthermore, we investigated the proliferation (see new **Supplemental Fig. 2a-c**) and want highlight the experimental setting for generation of the CAR33-KO NK cells: Two days after NK cell isolation, we performed the CAR33 transduction using always the same MOI of 2 for all different NK cell preparations. After one week of cultivation, we performed the nucleofection for CRISPR-based KLRC1 deletion using NK cells from the NT NK cells to generated KLRC1-KO NK cells and from the CAR33-NK cells, we generated CAR33-KLRC1-KO NK cell (see **Supplemental Fig. 2a**). Therefore, the further expanded CAR33 and CAR33-KO NK cells were generated from the same CAR expressing NK cells bulk. We observe a reduction in total cell number following nucleofection (**Fig. 1g**). However, once the engineered NK cells recover, the proliferation after nucleofection remains in the usual range. Therefore, we conclude that the reduction in cell number is due to the nucleofection method and seems to be not related to the specific KLRC1 deletion as NK cells treated with MOCK nucleofection (without RNPs) showed the same CFSE proliferation levels as CRISPR-KLRC1-KO NK cells (**Supplemental Fig. 2b-c**.) Finally, after few days of recovery the

expansion fold of KLRC1 and CAR33-KLRC1-KO NK cells is similar to the NT and CAR33 NK cells, see the gradient of the straight line in the updated **Fig. 1g**.

-Figure 2d: what is the comparator in this volcano plot? Is it CAR33 alone without KO? why would ZNF683 and CCL1 be upregulated and CCR2 be downregulated in this case in CAR33KO cells? This trend is opposite to what is seen in NTKO cells when compared to NT. How do the authors explain that?

Thank you for your question. We are fully aware that a proper introduction of the statistical model used in our analyses is essential for the reader to understand the presented results. We added the following information to the **Results** section in order to explain it better:

*"Our statistical model contained four variables and simultaneously estimated the inter-donor variability, the effect of CAR (Fig. 2b), the effect of KLRC1^{ko} (Fig. 2c) as well as the synergistic effect of the dual modification (CAR:KLRC1^{ko}, **Fig. 2d**). Of note, changes which the model accounts to the CAR:KLRC1^{ko} synergism are changes observed in the CAR33-KLRC1^{ko}-NK cells that cannot be explained by a simple combination of CAR and KLRC1^{ko} (**Fig. 2d**)."*

And:

*"Our computational method detected an interaction between CAR33 expression and KLRC1^{ko}, leading to the apparent upregulation of ZNF683 and chemokine (C-C motif) ligand 1 (CCL1) and downregulation of CCR2 (**Fig. 2d, Supplemental Fig. S3a**). This, however, could be explained by the fact that the changes induced by CAR33 and KLRC1^{ko} do not add up in the dual genetic modification: for example, both genetic modifications individually lead to downregulation of ZNF683, whereas the combination of both modifications does not further reduce ZNF683 expression."*

So, for the example of ZNF683 and CCL1 this would mean that in CAR-KO-NK cells ZNF683 and CCL1 are not downregulated as much as one would expect, if just adding up the effects of CAR33-introduction and KO of *KLRC1*. This can happen for example, if the gene is already completely downregulated by a single genetic modification and cannot be further downregulated. CCR2 is less upregulated in CAR-KO-NK cells than one might expect based on the combined effect CAR33-modified or *KLRC1*-KO NK cells. Again, a possible explanation is, that it already reached its maximal expression value after a single genetic modification. Overall, there are only slight changes in 3 genes which differ from the effects of each single modification. From that we conclude that the KO of *KLRC1* in CAR-NK cells constitutes a precise method of modification without inducing a major alteration of the cell state.

-Figures 2e and 2f are not well explained. Please elaborate regarding what new information they bring compared to the volcano plots. If no new information please move to supplementary.

We thank for the suggestion and comment. We agree, that the information of these two sub-figures can also be placed in the **Supplementary Figures** and adapted this accordingly in the revised manuscript. The added value from these data is that it displays not only the changes in expression, but also the absolute expression values. This modification allowed us to add exemplary protein expression and flow cytometry data to the revised version of **Fig. 2e, f**.

-Figure 3c: How were the 6 subclusters identified? No clear separation is seen in the tsne plots. Can the authors explain how did they delineate the subclusters?

Thank you for your question. We note that this comment is very similar to the comment raised by Reviewer Nr. 1 ("*CITE-seq on sorted NK cells allows the authors to identify six cell clusters. Although the authors define some clusters with features of more activated/mature NK cells, their identity is not clearly explained in the Results section.*").

We have copied the answer to the comment of reviewer 1 here:

"First, it is important to kindly keep in mind that the clusters identified in our analyses do not identify distinct subtypes of NK cells, but rather groups of NK cells with distinct transcriptional states. The reason why we perform clustering is that we want to highlight that the overall up-/downregulation of certain genes shown in **Fig. 2, 3 and Fig. 6** is due to a differential abundance of cells exhibiting a specific transcriptional state. The higher expression of certain groups of genes is restricted to specific subgroups of cells.

Regarding the computational methodology, we selected the parameters for clustering using different metrics. We added the following text to the methods section, to better explain how clustering was performed:

*"The normalized ADT and RNA feature counts were combined into one feature matrix. Principal component analysis (PCA) was performed with 20 components prior to Louvain clustering. Different values for the parameter k , which represents the number of neighbors, were tested and evaluated using clustering similarity scores (adjusted rand index (ARI) & weighted rand index (WRI)). The Louvain clustering result with $k=50$ returned a reasonable number of clusters and scored highest in both similarity scores (**Supplementary Fig. 3c**) and was used in downstream analysis. The results were projected on a t -distributed stochastic neighbor embedding (t -SNE)."*

As described in this addition to the methods section, a figure, which depicts the clustering behavior using different parameter values, is added into the **Supplementary Figure S3c**.

-Line 182-183: "Both KLRC1ko-NK cells and CAR33-KLRC1ko-NK cells were abundant in Cluster 4, which consists of NK cells expressing fewer activation markers." Isn't that against what the authors are trying to convey? the KO is making the NK cells less active? The panel the authors used seems to be T cell centric and not specific to NK cells. What about NK activation markers like DNAM1, NKG2D, NKG2C, Nkp44, Nkp46 and others?

We thank for the two comments. We have taken up the concerns and elaborated the cited phrase and statement more precisely and we also performed phenotypic analyses using a specific NK cell panel with typical NK cells markers (including CD16, DNAM1, NKG2D, Nkp44, Nkp46, CD69, TIGIT, Tim3, PD1).

In the revised version of the manuscript, we added to the phenotyping results of the NK cell profile to **Figure 2e, f**. In addition, we add the protein expression analysis of T cell markers, such as CD3, TCR α /b, CD4 to the **Supplementary Figure 3b**. In line with the CAR insertion, we observed a transcriptional RNA expression of TCR downstream signaling genes in the CAR33 NK cells independent of KLRC1 KO.

Accordingly, we have also added the following aspects to the **Results** and **Discussion section** (as highlighted): While no major phenotypic changes were observed in the different NK cell preparations (NT, CAR, KO, CAR-KO), there was an increase in activating CD69, NKG2D and DNAM1 in dual-modified NK cells after contact with the CD33+ target

cells, but also to some extent in PD1, TIM3 and TIGIT (**updated Figure 6j**). Ultimately, the balance between activating and inhibitory signals of each NK cell preparation will determine the resulting activation state of the genetically modified NK cells. As we understand it, the main advantage of the dual-modified NK cells is that they do not respond to the inhibitory NK cell checkpoint NKG2A / HLA-E axis, as the functional *in vitro* and *in vivo* tests have impressively shown (**Fig. 4,5,7**).

-Line 189-191: "To assess whether the reduction of NKG2A expression on the surface of CAR33-NK cells, which maintained their activated and mature state (Figure 2 and 3), would lead to increased cytotoxic capacity of CAR33-KLRC1ko-NK cells." This statement is not accurate since the authors already showed reduced activation in KO cells as evidenced by decrease in cluster 4.

Thank you, we agree with this comment. The reason why we come to the conclusion that these NK cells "maintained their activation and mature state" has been discussed in the comment above. We changed the text to: "*To assess whether the reduction of NKG2A expression on the surface of CAR33-NK cells, which "mostly" maintained their activated and mature state (Figure 2, 3 and 6), would lead to increased cytotoxic capacity of CAR33-KLRC1ko-NK cells.*"

-Figure 4c and associated supplementary videos: The data shown is misleading since the authors are not showing the important control which is CAR33 alone without KO.

Thank you for the comment. We fully agree with the remark and added all images showing all of the controls with CAR33 alone, NT and tumor only. Please see **updated Figure 4c** on the revised version of the manuscript.

-Figure 4d: Why do CAR NK cells show less cleaved caspase 3 compared to NT counterparts?

Thanks for the remark. We agree that the current presentation could be misleading and added additional labels to a clear and intuitive readability of **Figure 4d**. and we also refer to the additional donors showing the same results in the **Supplementary Figures (Suppl. Figure S4b)**.

Additional note: it is important to look at the smaller cleavage fragments below the thicker band around 20kDa. If they are present, it indicates that Caspase-3 was activated, and apoptosis is induced in this fraction of cells. This is mainly the case in CAR and CAR-KO-NK cells and not for NT-NK and KO-NK cells.

Please see Supplemental Figure S45: the Caspase-Data of tw additional donors show similar results.

-Figure 4e: Seems that the increase in granzyme B is due to the effect of the CAR and not the KO.

We thank for the comment. The reviewer is correct, and we stressed the part in the revised **Discussion**.

Briefly, we believe that in short-term cytotoxicity assays (4 hours) the CAR expression mediates the NK cell activation and fast killing capacity, while the KLRC1 KO plays an important role for long term cytotoxic activity due to removal of the inhibitory receptor of the HLA-E / NKG2A checkpoint. This is also evident in **Figure 4a**, where CAR NK cells perform better in **4h assays (top row)** compared to KO-NK cells and vice versa in **24h coculture assays (bottom row)**. CAR-KO-NK cells combine both features. We assume

that this combination of short and long-term activation of CAR-KO-NK cells leads to increased efficacy *in vivo*.

"Importantly, CAR33-KLRC1^{ko}-NK cells benefit from the additional modification, as they showed superior AML cell elimination in both short and overnight assays, especially at lower E:T-ratios, which reflects possible physiological conditions in a human. This combination of short- and long-term cytotoxicity appears to be mediated by the CAR-insertion, and maintaining functionality by removing the inhibitory receptor NKG2A may be crucial for NK cell efficacy in vivo."

-Line 213-216: "sorted NKG2A-, KLRC1ko- and CAR33-KLRC1ko-NK cells did not show increased killing of OCI-AML2 cells, compared to non-sorted KLRC1ko-NK cell population with approximately 50% NKG2A- cells, indicating that a 50%-KO of NKG2A was sufficient to lever out the inhibitory NKG2A-HLA-E axis." This might be true for an artificial in vitro assay but in vivo having residual NKG2A expression on the NK cells might still hinder their activity. This is especially true given the fact that IFN γ can upregulate HLA-E on tumor cells which can enhance NK cell inhibition in vivo.

We thank for the important comment. We fully agree with the argument and added further information to the **Discussion** part:

*"... However, while our work revealed that a 100% NKG2A-negative sorted CAR-NK cell population could not further increase cytotoxicity in vitro (**Supplementary Fig. S4c**), it remains to be clarified whether such a strong reduction in phenotypic NKG2A expression is necessary in the context of NK cell effector function, which is regulated by the balance of the activating and inhibitory cell surface receptor profile. In this regard, it is important to note that residual NKG2A expression might still hinder CAR-NK cell activity in vivo, due to potential upregulation of HLA-E on tumor cells following contact with IFN- γ secreted by NK cells."*

-Line 217-219: "PBMCs derived from three different AML patients with a substantial blast infiltration (50-95% CD45dim AML blasts) were co-cultivated for 4 hours with CAR33-KLRC1ko-NK cells." Why didnt the authors sort the blast populations and perform the killing assay? Interference from other cells in the PBMCs might muddy the results of the killing assay.

We understand this concern. As also mentioned to reviewer 1, in the revised version, we strengthen our data of AML primary blasts with samples from up to n=10 patients (low and high-risk molecular subsets). Please see the newly built **Figure 5**. In the first analyses of the former version of our manuscript, we included samples from n = 3 samples, with n = 1, which had only 50% AML blast. This one has been excluded from our final analyses, and we only included patient-derived samples with CD45dim AML blast > 90 % (see representative FACS data in **Figure 5e**). The killing was always confirmed with different NK cell preparations from n=22 healthy donors in total (see **Figure 5f, g**).

Of course, we also discussed sorting of the primary material, however, in our opinion, this might stress the leukemic cells and possibly reduce the viability of the patient material.

-Figure 5b: Why is CD4 being upregulated in the NK cells? Were these pure NK cells? is there any contamination with T cells?

We thank for the comment. We agree that this might be misleading as the analyses were done with pure NK cells. We have added a comment and a figure highlighting the evidence for NK cell purity: *"Both, the RNA and protein expression on single cell level confirmed the pure NK cell phenotype and absence of T cell markers (see **Suppl. Fig. 3**)."*

-CITEseq data in Figure 5d: It seems the effects are mediated by the CAR and the KO is not inducing transcriptional and phenotypic changes that would correlate with higher killing capacity. Do the authors think a longer co-culture time is needed? or phenotyping cells after exposure to tumor in vivo? The data as they are do not support the authors' hypothesis that there is synergy between NKG2A KO and CAR engineering.

We thank the reviewer for remark. We agree that the transcriptional effect induced by tumor cell contact is relatively similar and strong in both CAR and CAR-KO NK cells (**revised Fig. 6**). However, there is evidence for higher activation of CAR-KO NK cell after tumor cell encounter, consisting in higher abundance of cells in the transcriptional cluster 1 (**Fig. 6 d,e,f**).

In addition, as stated above, in the revised manuscript, we included phenotypic data after tumor exposure *in vitro* (**Fig. 6j**). As we understand it, the main advantage of the dual-modified NK cells is that they do not respond to the inhibitory NK cell checkpoint NKG2A / HLA-E axis, as the functional *in vitro* and *in vivo* tests have impressively shown (**Fig. 4,5,7**).

-Figure 6a: were the mice also injected with IL-2 or IL-15 to support NK in vivo proliferation and persistence?

Yes, the reviewer is right, the mice received daily IL-2 25000 U/mouse/day subcutaneously as shown in the revised scheme of **Figure 7a** and stated in **Methods**.

-Mice experiments:

The in vivo data are weak, only one mouse model was used, the time span of the experiment was short (mice sacrificed at 18 days), the authors used 2 mice only for survival analysis is not appropriate or sufficient to make any conclusions. These data need to be confirmed in a larger group of mice with at least 5 mice per group so that statistics can be performed. Using another mouse model will also strengthen the data. Isolating NK cells from the treated mice and comparing NKG2A KO vs NKG2A WT cells at the transcriptomic and proteomic level would be interesting since from the limited in vivo data it seems that NKG2A KO provides the CAR-NK cells with an advantage in vivo and the effect is more profound than what was seen in vitro.

-Figure 6d: No statistics, visually there doesn't seem to be significant differences between the different treatment groups. This does not correlate with the BLI images. The CAR33 seems to be performing worse than the other treatment groups based on these graphs.

-Figure 6e: The percentage numbers are so low even for the untreated mice, 2-3% in BM and 0.2-0.5% in spleen? This is odd given the strong BLI signal.

-Figure 6f: The results do not correlate with the BLI images where only CAR33 KO cells seem to be significantly controlling the tumor burden.

We thank the reviewer for the remarks and the comments. We address all comments with a **completely revised Figure 7** after additional mouse experiments and new additional **Supplementary Data** (all additions are highlighted in yellow the revised **Results** section). Since we agree with the reviewers' comments, we performed additional mouse experiments with survival analyses for the revision, and we are glad to confirm and underline the *in vivo* efficacy in this model.

Based on the comprehensive new data, we can address all the important remarks raised by the reviewer and we have taken the opportunity to completely redesign the former Figure 6, which has become **Figure 7**.

In brief, **Figure 7** includes a new additional and large mouse experiment with n=9 for CAR-KO treated mice (**Figure 7b,c**), BLI images confirming AML engraftment prior to treatment, additional survival data after CAR-KO treatment (**Figure 7d**), larger numbers of retransplant survival analyses (n=5/6) and MRD measuring (Suppl Figure 7).

-Lines 361-364: "CAR33-KLRC1ko-NK cells showed significantly increased bone marrow infiltration and cytotoxic capacity at lower cell numbers against CD33+/HLA-E+ OCI-AML2 cells compared to CAR33- or KLRC1ko-NK cells in vitro and in vivo without any signs of exhaustion or observable side effects." The authors did not provide data regarding functional exhaustion of the dual modified cells to support this claim. For example, performing long-term tumor rechallenge assays would be very insightful, or even prolonging the in vivo mouse experiments to see if the AML tumor cells eventually come back.

We thank the reviewer for the comment. Indeed, we were able to prolong the in vivo mouse experiments and added phenotypical data. Consequently, we we have adjusted our statement in the revised version of the manuscript.

As mentioned already above, please see the prolonged *in vivo* mouse experiment in the new **Figure 7b-d**. In our proof-of-concept experiments for dual modified CAR-NK cell therapy, we could show a significantly prolonged survival of CAR-NK cell treated mice to CAR and KO treated groups (**Figure 7d**). We also investigated the phenotype of CAR-KO NK cells upon tumor contact *in vitro* using a NK cell panel (see new version of **Figure 6j**). We see upregulation of CD69, DNAM1, TIGIT, TIM3, NKG2D and downregulation of NKp30, NKp46 in all gene-modified conditions with a major different in the CAR-KO cells.

Based on this new data we extended the discussion and adjusted the interpretation to the following:

"... CAR33-KLRC1^{ko}-NK cells showed significantly increased bone marrow infiltration and cytotoxic capacity at lower cell numbers against CD33⁺/HLA-E⁺ OCI-AML2 cells compared to CAR33- or KLRC1^{ko}-NK cells in vitro and in vivo without ~~any signs of exhaustion or observable side effects.~~"

-Lines 317-319: "Additionally, histological analysis of the liver, lung, and colon revealed no signs of GvHD or other therapy induced organ damages following double administration of CAR33-KLRC1ko-NK cells (data not shown)." Please show data, this would be important to ascertain safety of the dual modified cells.

We thank for the remark. We fully agree and in the revised version, we have added the requested histological analysis without any sign of inflammation or graft-versus-host disease. Please see new **Supplementary Figure S8**.

-Lines 413-414: "CAR33-KLRC1ko-NK cells predominantly eliminate AML cells using a Caspase 414 mediated killing mechanism, likely induced by the perforin/granzyme pathway." The data in Figure 4e show no difference in perforin and the data with increased granzyme b seem to be driven by the CAR.

We thank for this valuable comment, which complements the previous question addressing the role of granzyme. We agree that **Figure 4e** depicts a similar level of perforin, but increased granzyme B. As mentioned above, and we now provide a more differentiated revised description of the **Results** and **Discussion**:

Briefly, we believe that in short-term cytotoxicity assays (4 hours) the CAR expression mediates the NK cell activation and fast killing capacity, while the KLRC1 KO plays an

important role for long term cytotoxic activity due to removal of the inhibitory receptor of the HLA-E / NKG2A checkpoint. This is also evident in **Figure 4a**, where CAR NK cells perform better in **4h assays (top row)** compared to KO-NK cells and vice versa in **24h coculture assays (bottom row)**. CAR-KO-NK cells combine both features. We assume that this combination of short and long-term activation of CAR-KO-NK cells leads to increased efficacy *in vivo*.

"Importantly, CAR33-KLRC1^{ko}-NK cells benefit from the additional modification, as they showed superior AML cell elimination in both short and overnight assays, especially at lower E:T-ratios, which reflects possible physiological conditions in a human. This combination of short- and long-term cytotoxicity appears to be mediated by the CAR-insertion, and maintaining functionality by removing the inhibitory receptor NKG2A may be crucial for NK cell efficacy in vivo."

-Lines 415-417: "Overall, we demonstrated for the first time that CAR33-KLRC1ko-NK cells represent a novel strongly antileukemic NK cell phenotype, which evades exhaustion following AML contact, and appears to be perpetually activated, functional and safe in preclinical in vivo evaluation." Cannot claim that they are perpetually activated, the in vivo model was short, mice were sacrificed at day 18, so we dont really know what happens to the cells after prolonged exposure to AML in vivo and we dont know if they maintain functionality and do not show exhaustion through other mechanisms such as upregulation of TIIM3, LAG3, TIGIT or other inhibitory NK receptors.

We are very thankful for this comment. As mentioned above, we performed additional *in vivo* mouse experiments with repetitive applications of the different NK cell preparations (up to 3 x 5 Mio of NK cells / mouse), which allowed a prolonged clinical evaluation and survival analyses until > day 35 (**revised Figure 7a-d**). Additionally, we addressed the upregulation of inhibitory markers such as PD-1, TIM3 and TIGIT after longer co-culturing of NK cell preparations and tumor cells *in vitro* and could only observe moderate upregulation of several exhaustion markers but also activation markers, which may keep the balance towards functionality, but we are aware that this are *in vitro* data in a relatively artificial setting (**Figure 6j**). More strikingly, there was a clear reduction in leukemic progression in the longer animal experiments (new **Figure 7b**) and no leukemic outgrowth following bone marrow transfer from mice previously treated with the dual modified CAR33-KLRC1ko-NK cells (**Fig. 7h, j**).

In view of these additional data, but also the experimental limitations *in vitro* and in mouse models, we adjusted our claim: *"Overall, we demonstrated for the first time that CAR33-KLRC1ko-NK cells represent a novel strongly antileukemic NK cell treatment concept that can overcome AML-mediated immune suppression, and appears to be functional and safe in preclinical in vivo evaluation."*

-Many figures are missing statistics, please add that. For example: Figures 1c-f, 4e,f, 5g,h, 6b, 6-h

Thank you for the remark, and we apologize for these missing information. We now added statistics for the mentioned figures and additional (new) ones to underline the findings in the revised version of the manuscript.

Minor comments:

-Please number the figures in order. For example Figure 1e coming after figure 1g.

We fully agree with this remark. During the revision, we added an additional main and an additional supplementary figure and thus had the opportunity to redesign the layout according to this suggestion. We have carefully revised the manuscript in this respect.

-Figure 1a: please add the timeline of when CAR transduction and CRISPR gene editing were performed from the time of NK cell isolation. Please also indicate how NK cells were activated prior to modifications.

Thank you for this suggestion. We adapted **Figure 1a** accordingly and also adapted **Figure 1g** for a better understanding of the workflow.

-In the Introduction, please substitute “drugs” by “products” in the following sentence: “Research on novel chimeric antigen receptor (CAR)-T cell therapies led to the approval of multiple drugs for the treatment of B cell-derived malignancies.”

Thank you, we changed that accordingly.

-Line 159-160: “Genes encoding proteins expressed on immature or inhibited cells such as Selectin L (SELL) and Zinc Finger Protein 683 (ZNF683)” Please add references for this statement.

Thank you for the comment. We corrected this accordingly, and only mention ZNF683 since it was stated that it might negatively regulates INF-g production in NK cells (Post et al, Frontiers 2017, doi: 10.3389/fimmu.2017.00535). The reference was added to the text.

-No mention of use of cryopreserved NK cells in the results/figures but this is mentioned in the methods. Please clarify as cryopreservation has a deep impact on NK cell function and phenotype.

We thank the reviewer for this comment. We underlined the information in the Results, Figure Legends and Discussion of the revised version.

We fully agree that cryopreservation is an important issue also regarding clinical translation of “off-the shelf” NK cells products. If frozen products were used, we always performed quality control by functionality in cytotox assays (**Supplementary Figure 4d**). Also for CITE-seq analyses, we confirmed the cytotoxicity for the used donor 1 and donor 2 after thawing, with a superior functionality of CAR33-KLRC1 KO NK cells (**Supplementary Figure 4d**).

-What day after transduction were cells used for functional assays and in vivo assays? Please indicate that in the methods/results.

Thank you for the remark. We added this information to the **Methods** section:
“CAR33-KLRC1-KO-NK cells and control cells were cultivated for at least 10 days following nucleofection before being deployed in functional assays in vitro or in vivo.”
and
“NK cells were obtained from the same donor and had been cultivated for 18 days post nucleofection at the first time of injection and 25 days post nucleofection at the time of the second injection.”

Reviewer #3 (expert in acute myeloid leukaemia):

This is a well written manuscript in which the authors evaluate a dual modified (CD33 CAR transduced and KLRC1 ko) NK-cells in cell line and xenograft models and demonstrate enhanced anti-AML activity and a more mature and activated transcriptomic profile with such dual modifications that may suggest better efficacy with such NK-CARs in the clinical setting. I have the following critique / comments.

-The authors noted that the proliferation capacity of the KLRC1 KO CD33 NK-CAR was reduced compared with those without KLRC1 KO. Can they explain the reason for this phenomenon? This is somewhat concerning. Would partial KLRC1 KO be feasible and have continued desired effect of avoiding NKG2DA-HLA E axis inhibition while not compromising the NK-CAR proliferative potential?

We thank for the remark. We have investigated this issue further and found that the main reason is a transient reduction in cell numbers following electroporation and *not* the functional KLRC1 KO.

Please see also our reply to reviewer #1:

*"Furthermore, we want to highlight the experimental setting for generation of the CAR33-KO NK cells: Two days after NK cell isolation, we performed the CAR33 transduction using always the same MOI of 2 for all different NK cell preparations. After one week of cultivation, we performed the nucleofection for CRISPR-based KLRC1 deletion using NK cells from the NT NK cells to generate KLRC1-KO NK cells and from the CAR33-NK cells, we generated CAR33-KLRC1-KO NK cell. Therefore, the further expanded CAR33 and CAR33-KO NK cells were generated from the same CAR expressing NK cells bulk. We observe a reduction in total cell number following nucleofection (**Fig. 1g**). However, once the engineered NK cells recover, the proliferation after nucleofection remains in the usual range. Therefore, we conclude that the reduction in cell number is due to the nucleofection method and seems to be not related to the specific KLRC1 deletion as NK cells treated with MOCK nucleofection (without RNPs) showed the same CFSE proliferation levels as CRISPR-KLRC1-KO NK cells (**Supplemental Fig. 2b-c.**) Finally, after few days of recovery the expansion fold of KLRC1 and CAR33-KLRC1-KO NK cells is similar to the NT and CAR33 NK cells, see the gradient of the straight line in the updated **Fig. 1g.**"*

-In the co-culture experiment when PB blasts from 3 AML patients were co-cultivated for 4 hours with CAR33-KLRC1ko-NK cells please provide the baseline molecular and cytogenetic features as well as status (frontline or R/R) of these three AML populations. Do the authors postulate that a particular molecular or cytogenetic subset of AML would be more susceptible to CAR33-KLRC1ko-NK cell? Potentially based on known percent expression or MFI of CD33 or HLA-E in certain AML subsets? This would help select and enrich for certain populations in the initial phase I studies.

We thank the reviewer for this remark which is highly interesting and important in terms of clinical translation, and we are pleased to be able to provide all information requested. Importantly, during revision, we could increase the number of PB blast samples from AML patients (n=10) and addressed the molecular/cytogenetic subsets of these patients (**Supplementary Table 2**). Eventhough, we increased the number of patients and also the numbers of healthy NK cell donors (n=20) used in killing assays of these primary AML samples, we would need an even higher number of samples and functional readouts for correlation analyses. However, we could clearly confirm the increase in antileukemic cytotoxicity of the dual modified NK cell preparations against all primary AML samples so far, including 4 patients with standard risk profile and 6 patents with more complex karyotype and intermediate / high risk AML.

In regard to the comment on HLA-E expression: As mentioned also to reviewer #1, we performed expression analyses of HLA-E in a broad AML patient cohort (n=177) and confirmed that HLA-E is uniformly expressed in this clinical cohort, but did not correlate with clinical response (see **Figure 5** and **Supplementary Figure S5**).

Based on the updated new data, we conclude that the KO of *KLRC1* on CAR33 NK cells may be beneficial for the treatment of a majority of AML patients despite the specific AML subset classifications.

-The authors used the OCI-AML-xenograft model for *in vivo* assessment and showed that the CAR33-KLRC1ko-NK cells most effectively eradicated bone marrow and organ leukemia in mice. However these *in vivo* experiments could be much more representative if they were performed in patient derived xenografts encompassing various molecular subsets, especially high risk molecular subsets such as TP53m or inv3q or complex cytogenetics harboring AML. These are the populations of greatest unmet need and if the CAR33-KLRC1ko-NK cells show robust activity in such PDXs this could help focus on these populations in the initial phase I studies allowing for a more rapid and directed clinical development than trying to evaluated in all comer populations.

We agree that all these suggestions are very interesting. For the revision of this manuscript, we performed two additional mouse xenograft experiments *in vivo* to confirm all results with regard to the efficacy of dual modified CAR33-KLRC1ko-NK cells (as suggested by rev #2). Additionally, we could increase the number of primary AML patient-derived samples for further *in vitro* analyses (n=10) including high risk molecular AML subsets (**new additional Figure 5**).

We are fully aware that also patient derived xenograft would of high interest. However, especially for the suggested high risk material this will be challenging and would need a lot more time than we got for the extensive revision. Therefore, we hope to convince the reviewer by the possible clinical relevance of the genetically engineered CAR33-KLRC1ko-NK cells by the addition of the *in vivo* survival and retransplant analyses in the xenograft mouse model (please see the revised and re-arranged **Figure 7**).

-What do the authors anticipate will be the minimal effective number of CAR33-KLRC1ko-NK cells for dosing in patients with AML to start seeing efficacy. How frequent dosing would be needed?

We thank for the question and included a detailed comment to the entirely revised **Discussion** of the current version of the manuscript.

Since the data from the first in human Phase 1/2 clinical trial with CD19-CAR-NK cell preparations for treatment of B cell tumors has been reported by Katy Rezvani's group (Marin et al. Nature Medicine 2024), we have a clear idea, which dosing of a NK cell product is required for successful treatment of patients with hematological diseases. In line with this study, where dose escalation (n = 11) and dose expansion (n = 26) have been successfully performed for CAR NK cells, the range for a minimal effective dose for treatment of B-cell leukemia / lymphoma is 1×10^8 cells per kilogram of body weight, up to 10×10^8 cells per kilogram of body weight without severe side effects, such as neurotoxicity, cytokine release or graft versus host effects. Even if our dual modified CAR NK product is designed for treatment of AML, it would be reasonable to start with a similar dosing. Due to the advantage that CAR NK cells can be generated from allogeneic donors as an off the shelf product, it would be possible to address a weekly administration of CAR NK cells.

-While NK-cell based therapies have shown encouraging early efficacy in both lymphoma and AML the durability of responses has been questionable. What makes the authors believe this dual modified CAR product will have better durability. Is this something they could model in PDXs and demonstrate for example comparing such dual modified NK-cells versus traditional CD33 CART products and assessing not only AML burden clearance but time to relapse of AML in vivo models? Such data would be compelling to suggest the dual modifications implemented will not only help boost the initial cell kill but also potentially the durability of the NK response.

Thank you for the comment and all these suggestions for future follow-up investigations. Indeed, the direct comparison of CAR-T and CAR-NK cells in general is very interesting to elucidate advantages and disadvantages in terms of efficacy, survival and migration to identify the optimal CAR product.

However, the aim of our proof-of-concept study was to investigate whether we could further enhance the CD33-CAR efficiency (e.g. by Albinger et al, BCJ, 2022) by an additional KO of an inhibitory receptor. With that in mind, we designed this project. In the additionally performed mouse experiments – as described above - we could observe a significantly prolonged survival for mice treated with dual modified CAR NK cells compared to CAR only and KO only NK cells (**new Figure 7d**).

We are aware that our current success with the modified **CAR33-KLRC1ko-NK cells** mark just the beginning for durability of the response and further optimizations are needed and are possible due to the current technological advances in immune cell editing. In this direction, we plan an improvement of the CAR constructs by addition of IL-15 as reported in the Phase I CAR-NK cell clinical trial by Katy Rezvani's group (Marin et al. Nature Medicine 2024). We are happy to include these aspects into the **Discussion** part, with regard to future directions, but such additional experimental tasks are far beyond the scope of this manuscript.

- The discussion repeats much of the results and can be considerably shortened. Please avoid this and in the Discussion only focus on the implications of the key findings described in Results and future directions with this cell therapy approach.

We thank for the comment and fully agree with the suggestions from the reviewer. We hope the editor and reviewers will not only appreciate the additional amount of data and resulting conclusions, but also the entirely updated **Discussion**.

REVIEWERS' COMMENTS

Reviewer #1 (Remarks to the Author):

In the revised version of this manuscript, the authors addressed the majority of my concerns and my opinion is that the quality of this work is improved.

The authors did not confirm functional data using others AML cell lines, but they expand the experiments to a higher number of AML blasts. Overall, the additional experiments significantly strengthen their findings and CITE-seq clustering is discussed better.

I continue to feel that Materials and Methods section is too large. Information provided is important but some details could be moved in supplementary material.

Minor point:

- row 258: The authors report n = 10 different donor preparations. In contrast, Figure 5 represents n = 20 healthy donors.

Reviewer #2 (Remarks to the Author):

The authors did a fantastic job addressing all my comments and critiques and fixing all the deficiencies. The addition of the new in vivo data in Figure 7 have significantly strengthened the manuscript.

Could the authors please add the untreated and NT NK treated mice groups to the survival curves?

Otherwise I congratulate the authors for all their efforts at improving the manuscript.

[Note from Editor: This reviewer was also asked to comment on the Responses to Reviewer #3, who was no longer available to provide a review for the revised manuscript.]

I had a chance to review the rebuttal and manuscript in response to Reviewer 3. I think the authors did a good job responding to some of the queries. There were some experiments suggested by the reviewer that the authors did not perform but I agree with them that these experiments may be out of the scope of this paper. The only thing that I would suggest to the authors is citing prior literature showing that CAR-NK cells in combination with CRISPR KO modifications as a strategy has already been done by prior investigators: CISH KO CAR-NK cells (<https://doi.org/10.1182/blood.2020007748>), TIGIT KO CAR-NK cells (<https://doi.org/10.1016/j.omtm.2023.03.006>) among others.

Reviewer #1 (Remarks to the Author):

In the revised version of this manuscript, the authors addressed the majority of my concerns and my opinion is that the quality of this work is improved.

The authors did not confirm functional data using others AML cell lines, but they expand the experiments to a higher number of AML blasts. Overall, the additional experiments significantly strengthen their findings and CITE-seq clustering is discussed better.

I continue to feel that Materials and Methods section is too large. Information provided is important but some details could be moved in supplementary material.

Minor point:

- row 258: The authors report n = 10 different donor preparations. In contrast, Figure 5 represents n = 20 healthy donors.

Reply: We thank the reviewer for the feedback and we are glad to further improve our study accordingly. We addressed this minor final remark and corrected the n to n=20 donor preparations in row 248. Concerning the *Materials and Methods* section we proceeded as stated by the Editor and Publisher guidelines.

Reviewer #2 (Remarks to the Author):

The authors did a fantastic job addressing all my comments and critiques and fixing all the deficiencies. The addition of the new in vivo data in Figure 7 have significantly strengthened the manuscript.

Could the authors please add the untreated and NT NK treated mice groups to the survival curves?

Otherwise I congratulate the authors for all their efforts at improving the manuscript.

[Note from Editor: This reviewer was also asked to comment on the Responses to Reviewer #3, who was no longer available to provide a review for the revised manuscript.]

I had a chance to review the rebuttal and manuscript in response to Reviewer 3. I think the authors did a good job responding to some of the queries. There were some experiments suggested by the reviewer that the authors did not perform but I agree with them that these experiments may be out of the scope of this paper. The only thing that I would suggest to the authors is citing prior literature showing that CAR-NK cells in combination with CRISPR KO modifications as a strategy has already been done by prior investigators: CISH KO CAR-NK cells (<https://doi.org/10.1182/blood.2020007748>), TIGIT KO CAR-NK cells (<https://doi.org/10.1016/j.omtm.2023.03.006>) among others.

Reply: We thank the reviewer for the feedback and we are glad to improve our manuscript accordingly to these suggestions. Therefore, we added the data from all control UT and NT mice to the survival curves (see updated Figure 7d and 7i).

We thank the reviewer for the support and especially for the effort to additionally consider the comments of the previous reviewer #3 and to evaluate our work accordingly.

We included the requested literature and comments in the final discussion (see line 593-593 of the updated manuscript).